

# Beyond braid statistics: Constructing a lattice model for anyons with exchange statistics intrinsic to one dimension

Sebastian Nagies[1,2,3], Botao Wang[1,4], Adam C. Knapp[5],
André Eckardt[1*] and Nathan L. Harshman[6†]

**1** Technische Universität Berlin, Institut für Theoretische Physik,
Hardenbergstr. 36, 10623 Berlin
**2** Pitaevskii BEC Center and Department of Physics,
University of Trento, Via Sommarive 14,
**3** INFN-TIFPA, Trento Institute for Fundamental Physics and Applications, Trento, Italy
**4** CENOLI, Université Libre de Bruxelles, CP 231, Campus Plaine, B-1050 Brussels, Belgium
**5** University of Florida, Gainesville, FL 32610, USA
**6** Physics Department, American University, Washington, DC 20016, USA

⋆ eckardt@tu-berlin.de , † harshman@american.edu

## Abstract

Anyons obeying fractional exchange statistics arise naturally in two dimensions: Hard-core two-body constraints make the configuration space of particles not simply-connected. The braid group describes how topologically-inequivalent exchange paths can be associated to non-trivial geometric phases for abelian anyons. Braid-anyon exchange statistics can also be found in one dimension (1D), but this requires broken Galilean invariance to distinguish different ways for two anyons to exchange. However, recently it was shown that an alternative form of exchange statistics can occur in 1D because hard-core three-body constraints also make the configuration space not simply-connected. Instead of the braid group, the topology of exchange paths and their associated non-trivial geometric phases are described by the traid group. In this article we propose a first concrete model realizing this alternative form of anyonic exchange statistics. Starting from a bosonic lattice model that implements the desired geometric phases with number-dependent Peierls phases, we then define anyonic operators so that the kinetic energy term in the Hamiltonian becomes local and quadratic with respect to them. The ground-state of this traid-anyon-Hubbard model exhibits several indications of exchange statistics intermediate between bosons and fermions, as well as signs of emergent approximate Haldane exclusion statistics. The continuum limit results in a Galilean invariant Hamiltonian with eigenstates that correspond to previously constructed continuum wave functions for traid anyons. This provides not only an a-posteriori justification of our lattice model, but also shows that our construction serves as an intuitive approach to traid anyons, i.e. anyons intrinsic to 1D.

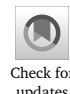



# Contents

# 1 Introduction

How do wave functions transform when indistinguishable particles are exchanged? In three dimensions and higher, this question has two answers: symmetric under pairwise exchanges like bosons or antisymmetric under pairwise exchanges like fermions. In either case, the transformation does not depend on the path taken by the exchange, only on the permutation. However in lower dimensions, the quasiparticles that emerge in interacting systems can possess different forms of exchange statistics. The explanation is topological: Unlike in three dimensions, particle interactions in one dimension (1D) and two dimensions (2D) create defects that make configuration space not simply connected. Different exchange paths for the same permutation can be topologically distinguished by how they wind around these defects. Anyons are particles with path-dependent exchange transformations, and they have statistics that lie intermediate between bosons and fermions.

Anyonic exchange statistics were first predicted for particles in 2D with hard-core two-body constraints [1,2]. Such interactions create defects in configuration space and the braid group describes how topologically-distinct exchange paths can wind around these defects [3–6]. Representations of the braid group determine how wave functions transform. The simplest abelian representations of the braid group give anyon wave functions with fractional exchange statistics described by a statistical angle $\theta \in [0, 2\pi)$ that interpolates between bosons $\theta = 0$ and fermions $\theta = \pi$. Two-dimensional anyons obeying fractional statistics can be realized as quasiparticles of topologically ordered states of matter, such as fractional quantum Hall states [7,8]. The anyonic braiding statistics with $\theta = 2\pi/3$ have been measured in the fractional quantum Hall effect through interference and scattering measurements [9,10].

In 1D, interactions are even more disruptive to the topology of configuration space; particles must pass through each other in order to exchange. Hard-core two-body interactions make exchange impossible in 1D, so implementations of fractional exchange statistics in 1D must find alternate methods to implement the required geometrical phases. The anyon-Hubbard model [11–13], which was recently implemented with ultracold atoms in optical lattices [14], uses number-dependent Peierls-phases to generate the required phases. These phases break spatial parity, time-reversal, and Galilean symmetry, and even in the continuum limit of the anyon-Hubbard model, these symmetries are not restored [15].

However, it was recently demonstrated that a different form of anyonic exchange statistics intrinsic to 1D is possible if the two-body hard-core relation is relaxed but a three-body hard-core constraint is enforced [16,17]. Hard-core three-body constraints in one-dimension also make the configuration space not simply-connected, and this allows for multi-valued wave functions and non-trivial exchange phases. Such a system is characterized by the so-called traid group which, like the braid group, can be visualized by strand diagrams with equivalence relations.[1] Similar to fractional statistics and the braid group, abelian traid statistics are also intermediate between bosons and fermions, but in a different manner. The traid group breaks the Yang-Baxter relations, so that pairwise exchanges of some neighboring particle pairs can be symmetric and antisymmetric for others, for instance in a staggered fashion with respect to the particle order (as is explained below and in Refs. [16,17]).

So far traid-anyonic exchange statistics has been studied only for quantum states in the continuum, where they are described by multi-valued wave functions. In this paper, we construct the first lattice model for traid anyons. For this purpose, we start from a one-dimensional bosonic lattice Hamiltonian with non-local, number-dependent tunneling phases. The phases embody the exchange properties of abelian representations of the traid group in a way reminiscent of bosonic descriptions of two-dimensional anyons via flux tube attachments and magnetic potentials. Using a non-local Jordan-Wigner-like gauge transformation, the bosonic model is then transformed to a local Hubbard-type Hamiltonian of anyonic particles obeying non-trivial, non-local commutation relations. The continuum limit of the bosonic representation of our model is found to give rise to solutions that correspond to the previously constructed traid anyonic wave functions, justifying our construction. We also find that, unlike the anyon-Hubbard model, the continuum limit of our traid-anyon-Hubbard model is Galilean invariant.

The motivation for our work is threefold: (i) We want to demonstrate that traid exchange statistics can be realized in a concrete model system. (ii) By directly engineering the required geometric phases associated with particle exchange, we want to provide an intuitive approach to the physics of traid anyons. (iii) Finally, we hope that our model might be a first step towards an experimental implementation of traid anyons, using techniques of quantum engineering in artificial quantum systems. While we do not propose a specific implementation for our model, we do identify the necessary structure of the number-dependent tunneling

---

[1]The strand group $T_N$ was independently discovered by several groups of mathematicians and goes by various other names, including the doodle group, planar braid group, and twin group [18–22].

phases that an implementation would require. Due to the non-local many-body interactions involved, a prospective implementation will presumably be more complicated than the recently-realized anyon-Hubbard model [14]. Apart from (i-iii), we also make the interesting observation that our model, which by construction describes particles with non-trivial traid-anyonic *exchange* statistics, also shows indications of approximate fractional Haldane-type *exclusion* statistics [23–25].

The paper is organized as follows: In Sec. 2, we review how non-trivial exchange statistics can be described by strand diagrams governed by rules that emerge from the topology of configuration space. In doing so, we compare abelian braid and traid anyons to particles obeying standard (bosonic or fermionic) statistics as well as to each other. In Sec. 3, we then construct the traid-anyon-Hubbard model by starting from a bosonic model with number-dependent Peierls phases. Numerical results presented in Sec. 4 then allow us to study the ground-state properties of the model. We find generalized Friedel oscillations as well as other indications for fractional exclusion statistics in both the dependence of the chemical potential on the total particle number and the occupations of the natural orbits. In Sec. 5 we finally take the continuum limit and show that in this limit the solutions of our model coincide with the previously predicted wave functions for traid anyons. The concluding section puts our work into a broader context and points towards future research.

## 2 Exchange statistics in continuum models

As a prelude to the lattice models in the next section, in this section we consider exchange statistics in continuum systems in first quantization. We first review the Symmetrization Postulate and how the symmetric group describes permutations of fermions and bosons. Then we show how topological defects created by hard-core interactions in low dimensions can break the relations of the symmetric group, leading to the braid group and traid group. The abelian representations of these groups provide multi-valued anyonic wave functions that, unlike the symmetric group, have path-dependent exchange transformations. These results can be rigorously derived using the so-called intrinsic approach to topological exchange statistics; see Appendix A for an overview and some references.

### 2.1 Standard statistics

Bosons and fermions, whether single-component or multi-component, have *standard* exchange statistics. Unlike anyons, there is no distinction required between particle permutations and particle exchanges; all exchange paths leading to the same permutation are equivalent. For standard exchange statistics, the Symmetrization Postulate is sufficient to describe wave functions of bosons and fermions. To implement the Symmetrization Postulate, bosons or fermions are given arbitrary particle labels, temporarily breaking indistinguishability. Then the artificial labels are made unobservable by symmetrizing or antisymmetrizing the wave function over these labels, effectively restoring indistinguishibility.

In more detail, consider $N$ particles with labels $i = 1, 2, \ldots, N$ and positions $x_i \in \mathcal{M}$ summarized in the vector $\mathbf{x} = (x_1, \ldots, x_N)$. When there are no points excluded by hard-core interactions then the configuration space is $\mathcal{X} = \mathcal{M}^N$, otherwise $\mathcal{X} \subset \mathcal{M}^N$. The permutations $p = \{p_1 p_2 \cdots p_N\}$ of these labels form the symmetric group $S_N$ and correspond to coordinate transformations on $\mathcal{X}$ that map the point $\mathbf{x} = (x_1, x_2, \ldots, x_N)$ to the point $\mathbf{x}' = p(\mathbf{x}) = (x_{p_1}, x_{p_2}, \ldots, x_{p_N})$. Any permutation $p \in S_N$ can be expressed as a sequence of pairwise exchanges $s_i$ with $i = 1, 2 \ldots, N - 1$, each of which swaps the particle labelled $i$ with the particle labelled $(i + 1)$. The transpositions $s_i$ form a generating set for the symmetric

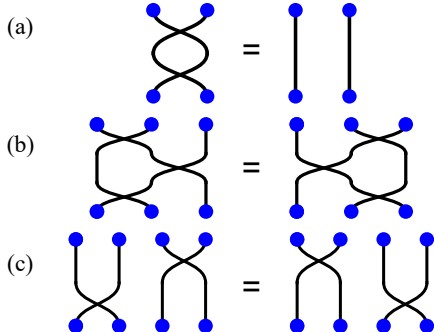

Figure 1: Examples of strand diagrams depicting equivalency relations (1). These are read from bottom to top, with particle 1 furthest left. (a) self-inverse $s_1^2 = 1$; (b) Yang-Baxter $s_1 s_2 s_1 = s_2 s_1 s_2$; (c) locality $s_1 s_3 = s_3 s_1$.

group and obey the relations

$$\text{Self-inverse: } s_i^2 = 1, \tag{1a}$$

$$\text{Yang-Baxter: } s_i s_{i+1} s_i = s_{i+1} s_i s_{i+1}, \tag{1b}$$

$$\text{Locality: } s_i s_j = s_j s_i, \text{ when } |i - j| > 1. \tag{1c}$$

A way to visualize these permutations is in terms of strand diagrams as shown in Fig. 1. Instead of a permutation of labels or a coordinate transformation on $\mathcal{X}$, the strand diagrams depict particle exchanges as $N$ continuous paths. The horizontal direction represents space and the vertical direction represents time (or a control parameter) increasing from bottom to top. The rules (1) can be understood intuitively as the equivalences achievable by continuously transforming the strands (moving and stretching in space, while maintaining forward progress in time) and allowing strands to cross and overlap arbitrarily. For the symmetric group, every strand diagram leading to the same permutation is equivalent under the rules (1); this is not true for braid and traid strand diagrams below.

Applying the structural rules (1) to first-quantized wave functions for $N$ indistinguishable particles gives only two possibilities: Bosons and fermions. To see this, consider the single-component (e.g., spinless or spin polarized) wave function $\psi(\mathbf{x})$ defined on the configuration space $\mathcal{X}$. The probability $|\psi(\mathbf{x})|^2$ to find particles at the positions $\mathbf{x}$ must be independent of the particle labels, so under the permutation $p$ of particle labels the wave function can only change by a phase factor $\eta_p$. Let the operator $\hat{U}$ be a representation of $S_N$ on the Hilbert space of wave functions on $\mathcal{X}$ defined by

$$\hat{U}_p \psi(\mathbf{x}) = \psi(p(\mathbf{x})) = \eta_p \psi(\mathbf{x}). \tag{2}$$

The phase factors $\eta_p$ provide an abelian representation of the symmetric group, so they must obey the constraints related to Eqs. 1. Denoting $\sigma_i$ as the phase associated with the permutation $s_i$, then an exchange of adjacent pairs gives $\hat{U}_{s_i} \psi(\mathbf{x}) = \sigma_i \psi(\mathbf{x})$. From Eq. (1a), we have that $\psi(s_i^2(\mathbf{x}))$ is equal to both $\sigma_i^2 \psi(\mathbf{x})$ and $\psi(\mathbf{x})$, implying that $\sigma_i = \pm 1$. In turn, from the Yang-Baxter relation (1b), we obtain $\psi(s_i s_{i+1} s_i(\mathbf{x})) = \sigma_i^2 \sigma_{i+1} \psi(\mathbf{x})$ is equal to $\psi(s_{i+1} s_i s_{i+1}(\mathbf{x})) = \sigma_i \sigma_{i+1}^2 \psi(\mathbf{x})$, implying that $\sigma_i = \sigma_{i+1}$ for all $i$. For abelian representations, locality (1c) provides no constraints on the phases. All together, we find that all phases are equal:

$$\sigma_i = \sigma, \quad \text{with} \quad \sigma = +1, \quad \text{or} \quad \sigma = -1, \tag{3}$$

where the two possible choices correspond to bosons and fermions, respectively.[2]

---

[2]Note that a similar argument also holds for multi-component wave functions but requires non-abelian representations of the symmetric group. For a detailed exposition on the Symmetrization Postulate in 1D, see [26, 27].

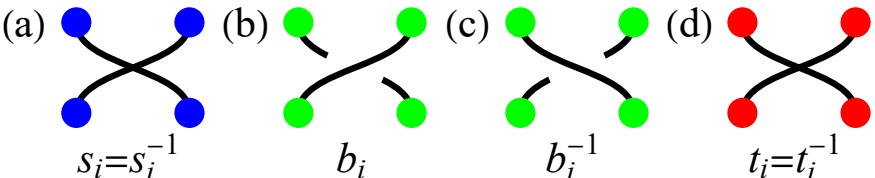

Figure 2: Strand diagrams depicting self-inverse relations for the symmetric group and traid group and the broken self-inverse relations for the braid group. Subfigure (a) and (d) depicts the self-inverse relations for $s_i$ and $t_i$, the pairwise exchange of particle $i$ and $i + 1$. Subfigures (b) and (c) depict the broken self-inverse relation for the pairwise exchanges $b_i \neq b_i^{-1}$. The two-body hard-core constraint means that strands cannot overlap, forcing distinction between pairwise exchanges that are right-handed $b_i$ and left-handed $b_i^{-1}$. If the particles live in three or more dimensions, these strands represent particle exchanges that could be continuously deformed into one another. In just one dimension, paths cannot go over or under, and so pairwise exchanges are topologically forbidden if there are hard-core two-body interactions. As we argue below, the anyon-Hubbard model allows to effectively define braid anyons also without a hard-core constraint in 1D.

## 2.2 Braid anyons

In 2D with hard-core two-body interactions, the equivalence between permutations of particle labels and particle exchanges no longer holds. Instead of the symmetric group $S_N$, particle exchanges for $N$ indistinguishable particles are described by the braid group $B_N$ [1, 2, 4, 5]. Because particles cannot coincide, the configuration space is not simply-connected and multivalued wave functions are allowed; see Appendix B for more details on how the braid group derives from topological exchange statistics.

The key structural features of the braid group can be understood by considering strand diagrams. Unlike the symmetric group, when two paths in the diagram cross, the particles cannot coincide and so we must indicate which particle is passing in front. As a result, the exchange of two particles is no longer self-inverse; the braids $b_i$ and $b_i^{-1}$ are the distinct over- and under-braided exchanges of the particles labeled $i$ and $i + 1$; see Fig. 2(b) and (c). With this distinction, the diagrams of Fig. 1 have to be modified as shown in Fig. 3. However, both the Yang-Baxter and locality relations are preserved because, unlike the self-inverse relation, continuous deformations of the strands described by those relations are not disrupted by the two-body hard-core constraints. Note that in three spatial dimensions also for hard-core interactions no distinction between over and under can be made since both diagrams can be transformed into each other using the third dimension. In one dimension, in turn, two-body hardcore interactions prevent any particle exchange. Thus, in the continuum, braid anyons are naturally expected to occur only in 2D. However, we show below how the discrete configuration space of a lattice allows for the definition of braid anyons also in 1D.

Every particle exchange $\gamma \in B_N$ can be represented as the product of $b_i$ and $b_i^{-1}$'s. These generators obey the relations

$$\text{Braid inequivalence: } b_i^2 \neq 1, \tag{4a}$$

$$\text{Yang-Baxter: } b_i b_{i+1} b_i = b_{i+1} b_i b_{i+1}, \tag{4b}$$

$$\text{Locality: } b_i b_j = b_j b_i, \text{ when } |i - j| > 1. \tag{4c}$$

Note that unlike the symmetric group, the same particle permutation can be enacted by multiple inequivalent exchange paths $\gamma \in B_N$. For example, the braids $b_1$, $b_1^{-1}$, $b_1^3$, $b_1^{-3}$, etc., all

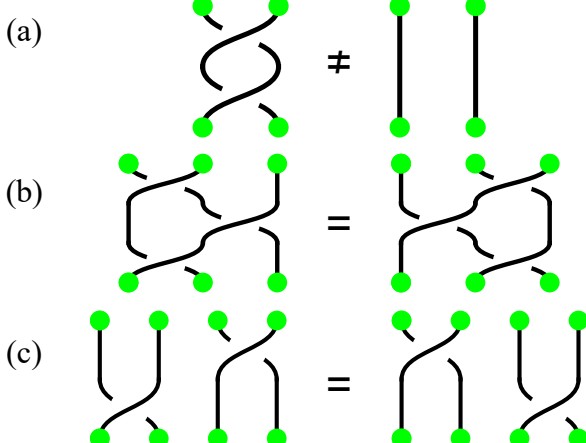

Figure 3: Strand diagrams depicting the braid group. Subfigure (a) depicts the consequence $b_1^2 \neq 1$ of the broken self-inverse relation $b_i \neq b_i^{-1}$. Subfigure (b) depicts the Yang-Baxter $b_1 b_2 b_1 = b_2 b_1 b_2$ relation and subfigure (c) depicts the locality relation $b_1 b_3 = b_3 b_1$, both unchanged from (1).

enact the same particle permutation of the first and second particles. The multi-valuedness of anyon wave functions on $\mathcal{X}$ originates from the multiplicity of topologically distinct exchange paths that execute the same permutation. Unlike the Symmetrization Postulate, elements of the braid group do not correspond to coordinate transformations on $\mathcal{X}$. However, we can define an operator $\hat{U}_\gamma$ that represents a particle exchange $\gamma \in B_N$, and for an abelian representation, $\hat{U}_{b_i}$ multiplies the wave function by a phase factor $\beta_i$:

$$\hat{U}_{b_i} \psi(\mathbf{x}) = \beta_i \psi(\mathbf{x}). \tag{5}$$

Because the generators are no longer self-inverse, the phase $\beta_i$ can take any value. As a result of the Yang-Baxter relation, as before we have $\beta_{i+1} = \beta_i$ and all the $\beta_i$ are required to be equal. Therefore, we obtain the continuous family of possible choices for abelian representations of $B_N$

$$\beta_i = e^{i\theta}, \quad \text{with} \quad \theta \in [0, 2\pi), \tag{6}$$

labeled by the exchange phase $\theta$. The special cases $\theta = 0$ and $\theta = \pi$ correspond to bosons and fermions, respectively, and lead to single-valued wave functions on $\mathcal{X}$. All other $\theta$ correspond to braid anyons with fractional exchange statistics and have wave functions that are multivalued on $\mathcal{X}$ [6].

## 2.3 Traid anyons

The equivalence between permutations of particle labels and particle exchanges also no longer holds when there are three-body hard-core interactions in 1D. The excluded points in configuration space are defects that result in topological exchange statistics described by the traid group $T_N$ [16, 17]. See App. C for how the traid group is derived from topological exchange statistics.

To understand the resulting exchange statistics, we can again modify the strand diagrams of Fig. 1, as shown in Fig. 4. Since we do not assume a two-body hard-core constraint, two strands can cross, just like in Fig. 1; there is no "over" or "under" distinction intrinsic to 1D, unless we encode this information in additional degrees of freedom like relative velocity. We can immediately verify that the exchange of particles is self-inverse and local. However, the Yang-Baxter relation is broken; there is no continuous deformation of the strand diagram on

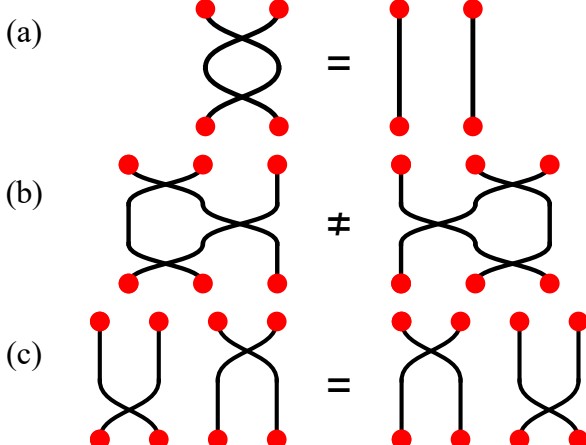

Figure 4: Strand diagrams depicting the traid group. Subfigures (a) and (c) depict the self-inverse $t_1^2 = 1$ and locality relations $t_1 t_3 = t_3 t_1$, both unchanged from (1). Subfigure (b) depicts the broken Yang-Baxter relation $t_1 t_2 t_1 \neq t_2 t_1 t_2$. The absence of a two-body hard-core constraint means that two strands can overlap, but the three-body hard-core constraint implies that three strands cannot. Therefore, a continuous deformation from the strand diagram on the left to the strand diagram on the right is impossible, and these two exchange paths are topologically distinguishable and may acquire different phases, despite leading to the same permutation.

the left-hand side of 1(b) to that on the right-hand side of 1(b). Such a deformation would necessarily involve crossing of three strands in one point, violating the three-body hard-core constraint.

As before, we can define generators $t_i$ for the exchange of the $i$th and $(i+1)$th particles that obey relations that we can read off from Fig. 4:

$$\text{Self-inverse: } t_i^2 = 1, \tag{7a}$$

$$\text{Traid inequivalence: } t_i t_{i+1} t_i \neq t_{i+1} t_i t_{i+1}, \tag{7b}$$

$$\text{Locality: } t_i t_j = t_j t_i, \text{ when } |i-j| > 1. \tag{7c}$$

With these relations, the $N-1$ generators $t_i$ define the traid group $T_N$ for $N$ strands. Note that while for the symmetric group the labels $i$ were an arbitrary assignment, and for the braid group the labels were based on an arbitrary positioning of the particles, in 1D there is a natural way to assign labels based on their ordering on the line; i.e. the particle furthest to the left (or equivalently, right) is the particle labeled $i = 1$, then the next particle in order is labeled $i = 2$, and so on. This ordering scheme is invariant under homeomorphisms of the line, and *ordinality* (so defined) is a topological property intrinsic to 1D systems. Therefore, unlike the pairwise exchange generators of $S_N$ or $B_N$, the generator $t_i$ exchanges *neighboring* particles in real space, not just in label space. This topological feature of ordinality is intimately connected with the breaking of the Yang-Baxter relation (7b).

Irreducible abelian representations of the traid group are specified by the phases $\tau_i$ picked up during the exchange of neighboring particles

$$\hat{U}_{t_i} \psi(\mathbf{x}) = \tau_i \psi(\mathbf{x}), \tag{8}$$

where the operator $\hat{U}_{t_i}$ represents the generator $t_i$. The self-inverse relation (7a) ensures that, like for standard $S_N$ statistics, $\tau_i = \pm 1$. However, unlike both standard and braid statistics, where the Yang-Baxter relation (7b) enforces equality of the phases, for the traid group the

phase factors $\tau_i$ are independent (generally, $\tau_i \neq \tau_j$). Thus, we have

$$\tau_i = +1, \quad \text{or} \quad \tau_i = -1, \tag{9}$$

leading to $2^{N-1}$ abelian representations of the traid group, corresponding to the $N-1$ choices of signs $\tau_i$. Like braid anyons, traid anyons contain bosons and fermions as special cases, here corresponding to all $\tau_i = +1$ or all $\tau_i = -1$, respectively. Unlike braid anyons, traid anyon representations do not continuously interpolate between these two extreme cases, but instead comprise a discrete set with non-standard exchange statistics.

To better understand traid anyon representations, consider the simplest non-trivial case of $N = 3$ particles. Besides bosons $(\tau_1, \tau_2) = (+1, +1) \equiv (++)$ and fermions $(--)$, there are two additional abelian representations $(+-)$ and $(-+)$. For the representation $(+-)$, the first pair of particles exchanges symmetrically like bosons and the second pair exchanges antisymmetrically like fermions. For either of these cases, the products $\tau_1 \tau_2 \tau_1$ and $\tau_2 \tau_1 \tau_2$ have opposite signs, showing that the same permutation of the first and third particle can be accomplished through topologically distinct exchange paths.

# 3 Lattice model for traid anyons

The path-dependence of exchange statistics in one-dimensional tight-binding lattice models is in some ways more intuitive and easier to visualize than the continuum case. The configuration space becomes a discrete Fock space and exchange paths become a series of discrete hops with dynamical phases. Exchange paths can be understood as loops in the discrete configuration space given by the Fock states, and the total dynamical phase can be visualized as a flux through this loop.

By engineering these phases and fluxes, bosonic lattice models can implement non-standard exchange statistics with density-dependent tunneling phases. In the first subsection we first recapitulate how this works for the anyon-Hubbard model, which was introduced to simulate fractional exchange statistics with bosons on a one-dimensional lattice [11]. In the next subsection, we reverse engineer the argument that led to the anyon-Hubbard model to create a Hubbard model with abelian traid-anyon statistics.

## 3.1 Anyon-Hubbard model with braid statistics

To introduce the anyon-Hubbard model, we follow Ref. [11] and begin from the deformed commutation relations hypothesized for fractional exchange statistics:

$$
\begin{aligned}
a_j a_k - a_k e^{i\theta \operatorname{sgn}(j-k)} a_j &= 0, \\
a_j a_k^\dagger - a_k^\dagger e^{-i\theta \operatorname{sgn}(j-k)} a_j &= \delta_{jk},
\end{aligned}
\tag{10}
$$

where $a_j$ is the annihilation operator for an anyon obeying fractional exchange statistics described by exchange phase $\theta$. In terms of these operators, the anyon-Hubbard Hamiltonian is then

$$H = -J \sum_j \left( a_{j+1}^\dagger a_j + \text{h.c.} \right) + \frac{U}{2} \sum_j n_j (n_j - 1), \tag{11}$$

where $n_j = a_j^\dagger a_j$ is the number operator on site $j$, $J$ is the tunneling strength, and $U$ is the two-body interaction strength.

The anyonic creation and annihilation operators with commutation relations (10) can be expressed in terms of bosonic ones $b_j$ via a generalized gauge transformation called the fractional Jordan-Wigner transformation:

$$a_j = e^{i\theta \sum_{k \leq j} n_k} b_j, \tag{12}$$

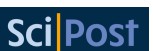

Figure 5: (a) Sketch of density-dependent hopping processes in the anyon-Hubbard model (Eq. 13). (b) Four configurations of two particles on three lattice sites. The parity-breaking density-dependent hopping phase means that the transition from the leftmost configuration to the rightmost configuration that takes place without coincidence accumulates a different phase than the lower path. Mapping this back to Fig. 3a, a clockwise loop returning to the same configuration corresponds to the braid generator $b_i$ and a counterclockwise to $b_i^{-1}$.

where also $n_j = b_j^\dagger b_j$. This density-dependent gauge transformation (12) is *non-local* because the phase for the operator $a_j$ depends on all occupation numbers to the left of site $j$. Although the fractional Jordan-Wigner gauge transformation is non-local, the deformed commutation relations (10) remain local.

Expressed in terms of boson operators, the anyon-Hubbard Hamiltonian (11) becomes

$$H = -J \sum_j \left( b_{j+1}^\dagger e^{-i\theta n_{j+1}} b_j + \text{h.c.} \right) + \frac{U}{2} \sum_j n_j (n_j - 1). \tag{13}$$

In the bosonic form, there are operator-valued, occupation-number-dependent Peierls phases attached to the tunneling matrix elements. These phases break parity, and they occur when a particle hops to the right onto an already occupied lattice site [Fig. 5(a)] or in the hermitian conjugate process when a particle hops to the left from a multiply occupied lattice site.

These Peierls phases can give rise to non-trivial geometric phases associated with closed paths in the many-body configuration space of indistinguishable particles, given by all Fock states with sharp site occupation numbers. Such geometric phases can be viewed as generalized magnetic fluxes. In Fig. 5(b), we show an example of a Fock-space loop associated with a nontrivial phase of $\theta$ ($-\theta$), when encircled in clockwise (anticlockwise) direction. Recently, the bosonic representation of the anyon-Hubbard model (13) was realized experimentally with ultracold atoms in an optical lattice, where the non-trivial geometric phases were probed directly via interference of the upper and the lower path of Fig. 5(b) during quantum walks [14].

The geometric phase resulting from the density-dependent tunneling can be directly related to the braiding of particles. In Fig. 5(b), we can associate paths that encircle the depicted loop, starting (say) from the leftmost configuration, with the exchange of the two particles. Encircling the loop in anticlockwise direction, so that the left particle tunnels rightwards twice, passing an occupied site, can be associated with $b_i$ depicted in the strand diagram Fig. 2(b), where the left particle moves over the right one. In turn, encircling the loop in clockwise direction, so that the right particle tunnels leftwards twice, passing an occupied site, can be associated with $b_i^{-1}$ depicted in the strand diagram Fig. 2(c), where the right particle moves over the left one. The corresponding phase factors are $\beta_i = \beta = e^{-i\theta}$ and $\beta_i^{-1} = \beta^* = e^{i\theta}$. In this sense, the anyon-Hubbard model describes braid anyons. Therefore, and in order to distinguish it from the traid-anyon-Hubbard model that we construct in the next subsection, we will also refer to it as the braid-anyon-Hubbard model in the following.

## 3.2 Hubbard model with traid exchange statistics

The goal of this section is to introduce a lattice model of interacting bosons that captures traid anyonic exchange statistics. For this purpose, we will reverse the strategy used for the braid-

anyon-Hubbard model above to implement these extreme cases. Namely, we first construct a bosonic model with non-local number-dependent tunneling phases that give rise to the desired exchange phases. Then we find the corresponding traid anyon field operators and their commutation relations by constructing a generalized Jordan-Wigner transformation, so that the kinetic energy term of the Hamiltonian becomes quadratic and local with respect to these new operators. We will first focus on the case of staggered abelian representations of the traid group, with alternating exchange phases given by $(+-+-\cdots)$ and $(-+-+\cdots)$. These are of specific interest, since they are furthest away both from the bosonic and (pseudo)fermionic cases, characterized by the homogeneous exchange phases $(++\cdots+)$ and $(--\cdots-)$, respectively. Subsequently, we will also construct a model for general abelian representations $(\tau_1, \tau_2, \ldots, \tau_{N-1})$ with $\tau_i = +1$ or $-1$ of the traid group.

A model that realizes the two abelian representations of the traid group with alternating exchange phases is achieved by the following lattice model using bosons with number-dependent hopping phases,

$$H = -J \sum_{j}^{L} \left( b_{j+1}^{\dagger} e^{i\pi(N_j+I)n_{j+1}} b_j + \text{h.c.} \right) + \frac{U}{2} \sum_{j}^{L} n_j(n_j - 1). \tag{14}$$

In contrast to the braid-anyon-Hubbard model (13), the number-dependent hopping phase is now defined in terms of the operator

$$N_j \equiv \sum_{k \leq j} n_k, \tag{15}$$

that counts the number of particles to the left of (and including) lattice site $j$. The integer $I \in \{0, 1\}$ that appears in the phase determines which of the two possible abelian representations of the traid group with alternating exchange phases the model realizes: For $I = 0$ we have $(+-+-+\cdots)$, meaning that the two leftmost particles exchange like bosons, the second and third particles from the left exchange like fermions, and so on. For $I = 1$ on the other hand, the above pattern gets flipped to $(-+-+\cdots)$, i.e. the two leftmost particles now exchange like fermions.

The symmetries of the lattice traid anyon model can be inferred from the bosonic form of the Hamiltonian (14). Time reversal invariance $\mathcal{T}H\mathcal{T} = H$ results by observing that $\exp(i\pi k) = \exp(-i\pi k)$ for any integer $k$, so the Peierls phase is invariant under complex conjugation, as are all other terms. Spatial inversion $\mathcal{P}$ transforms $j$ to $L - j$ and $N_j$ to $N - N_j$. After reindexing the sum, the only difference between $\mathcal{P}H\mathcal{P}$ and $H$ that remains is an additional phase $\exp(i\pi N)$ in the hopping terms. Therefore, for $N$ even, we have $\mathcal{P}H(I)\mathcal{P} = H(I)$ with the same $I$ whereas for $N$ odd we have $\mathcal{P}H(I = 0)\mathcal{P} = H(I = 1)$. In comparison, neither $\mathcal{T}$ nor $\mathcal{P}$ are symmetries of the braid-anyon-Hubbard model, although a gauge-transformed combination of them is [28].

Figs. 6(a-c) depict these alternating hopping phases $(+-+)$ for the case of 4 particles and $I = 0$: When the leftmost particle hops to the right onto an already occupied lattice site [Fig. 6(a)], a phase factor of $+1$ gets picked up. Tunneling to the right onto an empty site always gives a trivial phase factor, and so if that particle were to then hop onto another empty site (not depicted), the two particles would be exchanged in order with an overall exchange phase factor of $+1$. Exchanging the third and fourth particle gives the same result, while a swap of the central two particles [Fig. 6(c)] results in an overall phase factor of $-1$, i.e. the second and third particles exchange like fermions. Figs. 6 (d,e) show how these phases accumulate during particle exchanges. We can see that, like for the braid-anyon-Hubbard model, the number-dependent Peierls phases give rise to the desired exchange phases.

Note that our lattice model in Eq. 14 uses bosons and thus allows an arbitrary number of particles to occupy the same lattice site. However, a three-body hard-core constraint is essential

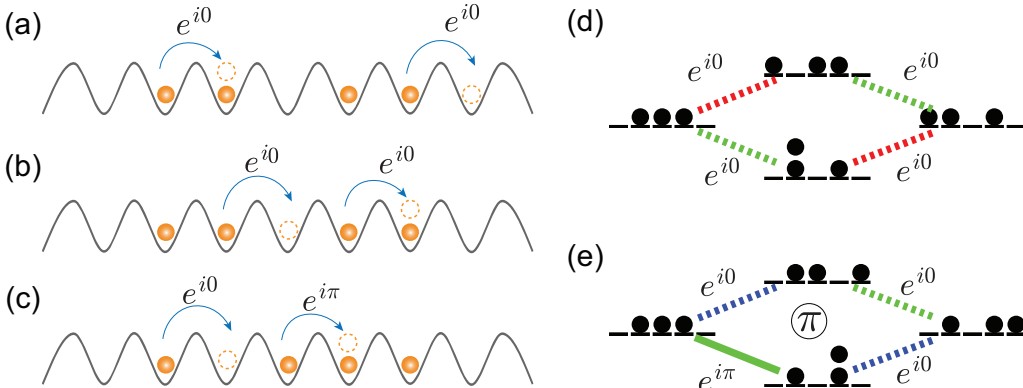

Figure 6: (a-c) Sketch of some possible hopping processes and associated phase factors in our lattice model for traid anyons (Eq. 14). Depicted is a system of 4 particles with an integer $I = 0$, corresponding to the abelian representation $(+-+)$. (d-e) Two loops in configuration space for the $N = 3$ traid anyon model representation $(+-)$, analogous to Fig. 5. Unlike the braid-anyon-Hubbard model, all hopping parameters are real, so there is no difference between clockwise and counterclockwise loops, i.e., these exchanges are self-inverse. In (d), the first and second particles undergo an exchange process. All hopping phases have the same sign and the product of all the phase factors is $+1$. In (e), the second and third particles undergo an exchange process. The sign of the lower left hop is reversed, so the product of all the phase factors is $-1$.

for the existence of genuine traid anyons in the continuum. Indeed, if we consider hopping processes on the lattice involving three or more particles, the desired pattern of alternating exchange phases breaks down for our model. We can address this problem theoretically in two ways: We can add a three-body interaction term to the Hamiltonian (14) and take the limit of the coupling coefficient to infinity, or we can implement the three-body hard-core constraint algebraically by requiring $b_j^3 = 0$. For numerical convenience, we have chosen the latter approach by simply truncating the state space so that at most two particles can occupy the same site. Note that if we restrict ourselves to the low-energy, dilute limit, then three-body occupation of the same site becomes increasingly unlikely to occur and should be negligible for the calculation of most observables. Our numerical results in the next section show that this is indeed the case for the low-energy dilute limit. Further, in the following section on the continuum limit we show that three-body, hard-core interactions emerge naturally from the number-dependent tunneling phases for the alternating representations.

We now define annihilation and creation operators, $t_j$ and $t_j^\dagger$, for the traid anyons. For that purpose, we define a generalized Jordan-Wigner transformation, so that the kinetic part of the Hamiltonian describing tunneling becomes local and quadratic with respect to these new operators:

$$H = -J \sum_j^L \left( t_{j+1}^\dagger t_j + \text{h.c.} \right) + \frac{U}{2} \sum_j^L n_j (n_j - 1). \tag{16}$$

For the staggered traid anyon model (14), this is achieved by:

$$t_j \equiv e^{-i\pi M_j} b_j, \tag{17a}$$

$$M_j \equiv \frac{1}{2} \left[ \sum_{k \leq j} n_k \left( N_j - n_k \right) \right] + I N_j, \tag{17b}$$

where $M_j$ depends on the $N_j$ (15) and on the integer $I$ (determining which of the two representations with staggered exchange phases is realized in Eq. 14). Like the braid-anyon-Hubbard model, the transformation (17) leaves the number operator on site $j$ invariant

$$t_j^\dagger t_j = b_j^\dagger b_j = n_j \, . \tag{18}$$

The crucial difference is that when expressed in the bosonic form, the hopping phases of the traid anyon model (14) are non-local and depend on the notion of ordinality, whereas hopping phases for the braid-anyon-Hubbard model are local.

This non-local property is noticeable also in the commutation relations for the traid operators $t_j$. From Eqs. 17 and the bosonic commutation relations, we obtain:

$$t_j t_k - t_k e^{i\pi \mathrm{sgn}(j-k)\left[N_{jk}+I\right]} t_j = 0 \, , \tag{19a}$$

$$t_j t_k^\dagger - t_k^\dagger e^{-i\pi \mathrm{sgn}(j-k)\left[N_{jk}+I\right]} t_j = \delta_{jk} \, , \tag{19b}$$

where $N_{jk}$ are new operators defined as

$$N_{jk} \equiv \begin{cases} N_j - n_k \, , & \text{if } j \geq k \, , \\ N_k - n_j \, , & \text{if } j < k \, . \end{cases} \tag{20}$$

From Eqs. 19, we see that the commutation relations of the traid anyon operators reduce to the bosonic case when acting on the same lattice site. However, for operators acting on different sites, the commutation relations are either fermionic or bosonic, depending on the configuration of all the other particles on the lattice.

The traid anyon lattice model (14) generalizes to arbitrary abelian representations of the traid group in the following way:

$$H = -J \sum_j^L \left( b_{j+1}^\dagger e^{i\pi F(N_j)n_{j+1}} b_j + h.c. \right) + \frac{U}{2} \sum_j^L n_j(n_j - 1) \, , \tag{21}$$

where $F(N_j)$ is a polynomial of $N_j$ (15) that takes on even or odd integer values for integer inputs. $F(N_j)$ differs for each abelian representation and takes on the simplest form for the alternating cases: $F_{\mathrm{alt}}(N_j) = N_j + I$. For example, if one wanted to construct a model to realize the representation $(++-)$ for 4 traid anyons, we would need to find a polynomial that takes on the values $F_{++-}(0) = 0$, $F_{++-}(1) = 0$ and $F_{++-}(2) = 1$ (or equivalently other even/odd integer values). In practice, for our numerical simulations in Sec. 4, we do not construct the polynomial explicitly and only set the desired inputs/outputs.

Analogously, we can find a Jordan-Wigner-like transformation that brings the generalized Hamiltonian of the traid anyon model (21) to the form (11) with a quadratic nearest neighbor tunneling term:

$$t_j \equiv e^{-i\pi f(N_j)} b_j \, , \tag{22}$$

where $f(N_j)$ is a polynomial of $N_j$ (15), taking on odd or even integer values depending on the traid group representation which is to be realized. In particular, $f(N_{j+1}) - f(N_j) = F(N_j)n_{j+1}$ should be fulfilled, so that the kinetic term in Eq. 21 is quadratic when expressed in terms of these general traid anyon operators. For example, to construct traid anyon operators for the representation $(++-)$, we can set $f(0) = f(1) = f(2) = 0$ and $f(3) = 1$. It is easy to check that this yields the desired pattern of tunneling phases, provided we exclude three-body coincidences. Similarly, the generalized transformation (22) leads to generalized commutations relations for the $t_j$.

We note that the transformation defined in Eqs. 17 for the alternating abelian representations does not match the general form given by Eq. 22: The operator $M_j$ is not just a polynomial

of $N_j$ but also depends on the configuration of the particles to the left of site $j$ (and not just their number). For instance, a two-body coincidence to the left of site $j$ would result in an extra factor of $-1$ in $e^{-i\pi M_j}$. These potential additional factors of $-1$ cancel out for terms of the form $t_{j+1}^\dagger t_j$ or $n_j = t_j^\dagger t_j$ and have thus no influence on the Hamiltonian in Eq. 16. On the other hand, for observables like the two-point correlation function $\langle t_i^\dagger t_j \rangle$, it can indeed make a difference, e.g. if there is a two-body coincidence between lattice sites $i$ and $j$. The factor of $-1$ then only occurs for one of the traid anyon operators and does not cancel out. The advantage of using the traid anyon operators defined in Eqs. 17 for the alternating abelian representations instead of the general form given by Eq. 22, is that the transformation to boson operators always results in the conceptually simple Hamiltonian in Eq. 14, even if we do not enforce a three-body hard-core constraint.

Finally, we would like to point out that the representation of traid anyons using bosons with number-dependent tunneling matrix elements resembles the composite-particle picture of braid anyons in 2D (see, e.g. Ref. [6]). In the latter case, 2D braid anyons are mapped to charged bosons (or fermions) to which a magnetic flux is attached to implement the geometric phases picked up when particles are moved around each other. In the former case of our traid anyon lattice model, the number-dependent tunneling matrix elements also give rise to generalized magnetic fluxes in the discrete configuration space of our system that are determined by the position of the particles and implement the geometric phases associated with the exchange of traid anyons; [See e.g. Figs. 5(b) and 6(e)].

# 4 Ground-state properties and signatures of fractional exclusion statistics

After introducing our lattice model for traid anyons above, we investigate its ground state properties in this section. To that end we numerically calculate the particle densities, energies and natural orbitals via exact diagonalization of Eqs. 14 and 21 (supplemented by DMRG simulations for higher filling factors in Fig. 9). These results show how the abelian traid group representations lie between fermions and bosons, and also hint at an intriguing connection to Haldane's fractional exclusion statistics [23] (c.f. App. D).

For all our simulations in this section, we set the on-site interactions to zero ($U = 0$). We also enforced a three-body hard-core constraint to reduce computation times and to always preserve the characteristic pattern of exchange phases, which breaks down for tunneling processes involving three or more particles in our traid anyon lattice model (14). Because we stay in the low-energy, dilute regime for most of our simulations, three-body processes can be neglected for the calculated observables (see also the discussion in section 3.2). Consequently, the signatures of traid anyons reported below do not rely on this additional constraint.

## 4.1 Particle density: Generalized Friedel oscillations

As a first physical observable, we consider the ground state particle densities. The particle density is independent of whether we formulate the model in terms of boson or traid anyon operators: $\langle n_j \rangle = \langle b_j^\dagger b_j \rangle = \langle t_j^\dagger t_j \rangle$, i.e. the number of bosons and traid anyons on each lattice site is identical. Fig. 7 shows the ground state densities of 3 to 6 traid anyons on 20 lattice sites. We depict the two possible abelian representations with alternating exchange phases $(+-+-...)$ and $(-+-+...)$ for each number of particles. As expected, the reflection symmetry of the densities about the midpoint of the lattice embodies the symmetry of the arrangement of the exchange phases: While the individual density profiles for the representations with $I = 0$ and $I = 1$ are not symmetric for odd $N$, they are mirror images of each other. In turn for even particle numbers, we find distinct, individually symmetric densities.

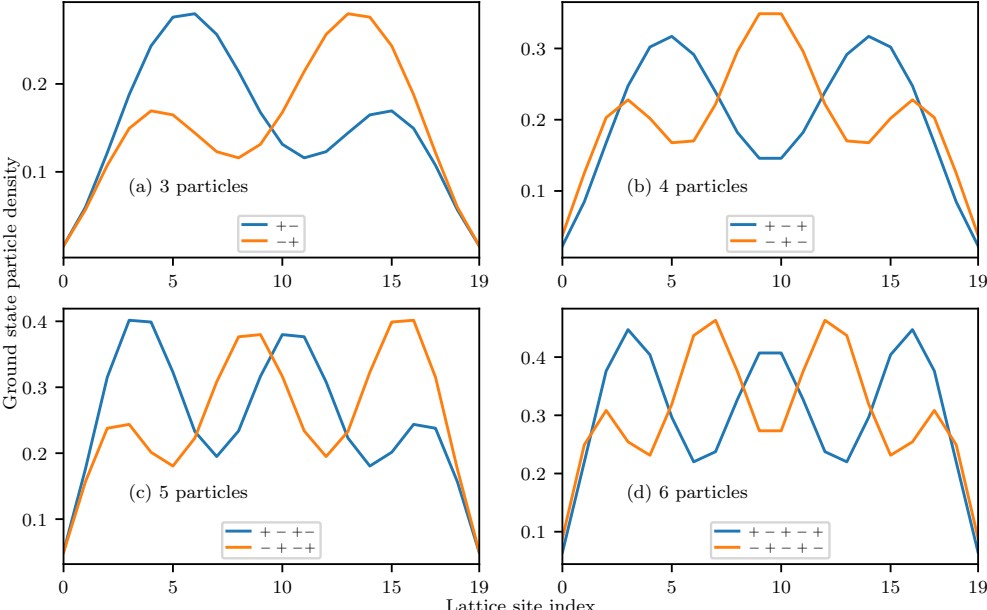

Figure 7: Ground state particle densities for the traid anyon lattice model (14) for 3-6 particles on 20 lattice sites and $U = 0$. Each plot shows the densities corresponding to the two possible abelian representations of the traid group with alternating exchange phases of a given number of particles, i.e. the blue lines correspond to the $I = 0$ model where the first and second particles exchange like bosons, the second and third like fermions and so on. The exchange phases for $I = 1$ are flipped.

Examples of non-alternating abelian representations of the traid group are shown in Fig. 8, where the densities of all 8 possible abelian representations (not just the alternating cases) for 4 particles are represented. The top left plot depicts the trivial case of bosons and pseudo-fermions, where all exchange phases are identical. The bottom two plots show 4 representations which can only be realized by the generalized version of our lattice model (21).

The density profiles depicted in Figs. 7 and 8 show that for each minus sign in the chosen traid-group representation, there is a minimum separating two local density maxima. Intuitively, depending on the exchange phases on both sides of the minus sign, these maxima are associated with either single particles resembling fermions, or two particles forming a 'bosonic' pair (separated by fermionic exchanges from its neighbors). For non-alternating traid group representations, when there are several plus signs next to each other, also 'bosonic' triples, quadruples, etc. can be observed (see Fig. 8). In the case of pseudo-fermions, i.e. the representations $(---\cdots)$ with only minus signs (see the top left plot in Fig. 8), these oscillations of the density correspond to Friedel oscillations [29], as they have been predicted also for the braid-anyon-Hubbard model [13]. These oscillations are a result of the fermionic two-particle correlations (reflecting Pauli exclusion), as they become visible in the density distribution close to a local defect, which is here given by the edge of our finite system with open boundary conditions. Thus, for the non-trivial representations, containing both plus and minus signs, the observed oscillations can be viewed as generalized Friedel oscillations.

## 4.2 Ground state energy and chemical potential

Like the particle densities discussed above, the ground-state energy (as well as the full spectrum) does not depend on whether we consider bosons with number-dependent tunneling

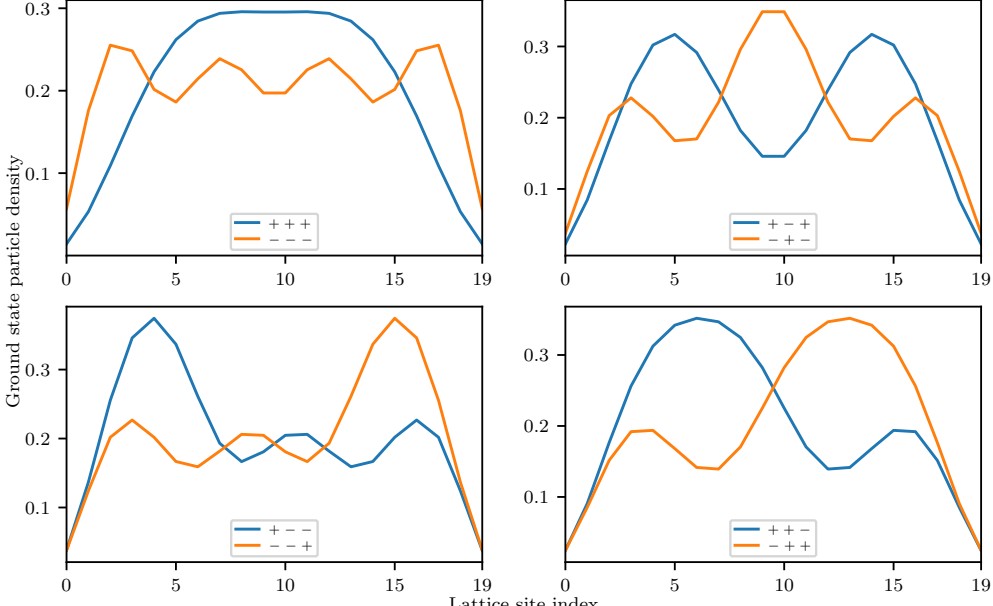

Figure 8: Ground state particle densities for the generalized traid anyon lattice model (21) on 20 lattice sites for $U = 0$. Shown are all 8 possible abelian representations of the traid group for a system of 4 particles.

phases or traid anyons, since the Hamiltonians for both pictures are equivalent due to the construction of the transformation between them (17). We numerically calculate the ground state energies $E_0$ of our lattice model and consider the zero-temperature chemical potential, i.e. the ground state energy difference $E_0(N) - E_0(N-1)$ of a system with $N$ and $N-1$ respective particles. Note that this quantity is well defined only for a specific rule for how the sign $\tau_N$ is chosen, when the $(N+1)$st particle is added to the system.

The chemical potential directly provides a perspective on the relation between exchange statistics and exclusion statistics (see Appendix D below). Free bosons and free fermions (where by "free" we mean non-interacting) are the extreme cases. Any number of bosons can occupy the ground state, so each additional particle requires the same amount of energy, whereas each additional fermion must occupy the next higher single-particle energy eigenstate. Free particles with fractional exclusions statistics, where single-particle states can be occupied by a well defined finite number of particles, would, in a similar fashion, give rise to a clear fingerprint in the behaviour of the chemical potential as a function of the total particle number. Below, we will, indeed, see traces of such behaviour, but also deviations from it, which will be attributed to the fact that, as a result of the number-dependent tunneling phases, even for $U = 0$ the traid-anyon Hubbard model does not describe free particles.

We show the results for a system with 30 lattice sites, up to 10 particles and no on-site interactions ($U = 0$) in Fig. 9. We compare the two realizations of traid anyons with alternating exchange phases (14) to (spinless) fermions, pseudo-fermions, bosons, and to the braid-anyon-Hubbard model (13) with exchange phase $\theta = \frac{\pi}{2}$.

The ground state energies of free fermions and free bosons can be calculated analytically from the single-particle energies $E(k) = -2J \cos(dk)$ of a system with $M$ lattice sites, lattice spacing $d$, and allowed quasi-momenta $k = \frac{\pi \nu}{d(M+1)}$ with $\nu = 1, 2, ..., M$. For the bosonic ground state all bosons occupy the single-particle ground-state, thus $E_0(N) - E_0(N-1) = -2J \cos(\pi/(M+1))$. For (spinless) fermions on the other hand, the Pauli exclusion principle implies that $E_0(N) - E_0(N-1) = -2J \cos(N\pi/(M+1))$. The ground state energies of the traid-

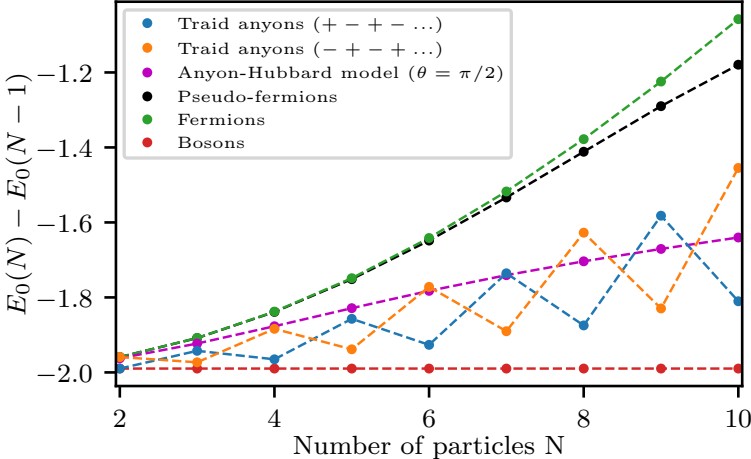

Figure 9: Change in ground state energy (chemical potential) in units of $J$ after adding the $N$th particle to a system with $N-1$ particles on 30 lattice sites and no on-site interaction ($U = 0$). Traid anyons (14) with alternating exchange phases $(+-+-...)$ or $(-+-+...)$ are compared to fermions, pseudo-fermions, bosons, and the braid-anyon-Hubbard model with $\theta = \frac{\pi}{2}$. Except for bosons and fermions, a maximum site occupation of 2 particles was enforced.

anyon-Hubbard and braid-anyon-Hubbard model (13) were obtained numerically using exact diagonalization for $N < 7$ and DMRG for $N \geq 7$. For completeness, we have also included pseudo-fermions corresponding to the $(--\cdots-)$ representation of the traid-anyon Hubbard model as well as to the braid-anyon-Hubbard model with $\theta = \pi$. Their chemical potential agrees nicely with that of free fermions in the dilute regime (roughly below quarter filling or $N = 7$), but as the density increases one can see a deviation because multiply-occupied sites are allowed for pseudo-fermions, unlike for true fermions.

We chose the braid-anyon-Hubbard model with exchange phase $\theta = \frac{\pi}{2}$ for comparison, because it provides a different notion of particles exhibiting behaviour 'right in the middle' between fermions and bosons: The anyons in the braid-anyon-Hubbard model always exchange with the same phase $\theta = \frac{\pi}{2}$, averaging between bosonic and fermionic behavior. In contrast, traid anyons in the representations $(+-+-...)$ or $(-+-+...)$ alternate between exchanging fermion-like (corresponding to an exchange phase $\theta = \pi$) and boson-like ($\theta = 0$). These different notions of interpolation between fermions and bosons are also reflected in the chemical potentials in Fig. 9. While the chemical potential lies approximately in the middle of the fermionic and bosonic case for both traid anyons and the braid-anyon-Hubbard model with $\theta = \frac{\pi}{2}$, the behaviour is nonetheless strikingly different: We observe a smooth increase of the potential for the braid-anyon-Hubbard model, while the traid anyons exhibit a step-like behaviour, where the change in the chemical potential increases or decreases in an alternating fashion.

The step-wise increase of the chemical potential, observed for the staggered traid anyons, can be interpreted as a signature of approximate Haldane-like *semionic* fractional exclusion statistics, i.e. of a generalized Pauli exclusion principle, where single-particle states can be occupied by up to two particles (c.f. App. D). Ideally, such behaviour might be associated with the chemical potential increasing for every other particle added and staying constant otherwise. We, however, find that rather than staying constant in the latter case, the chemical potential decreases. This hints at an effectively induced attractive interaction, something previously noted for the braid-anyon-Hubbard model [30]. In the next subsection, where we investigate

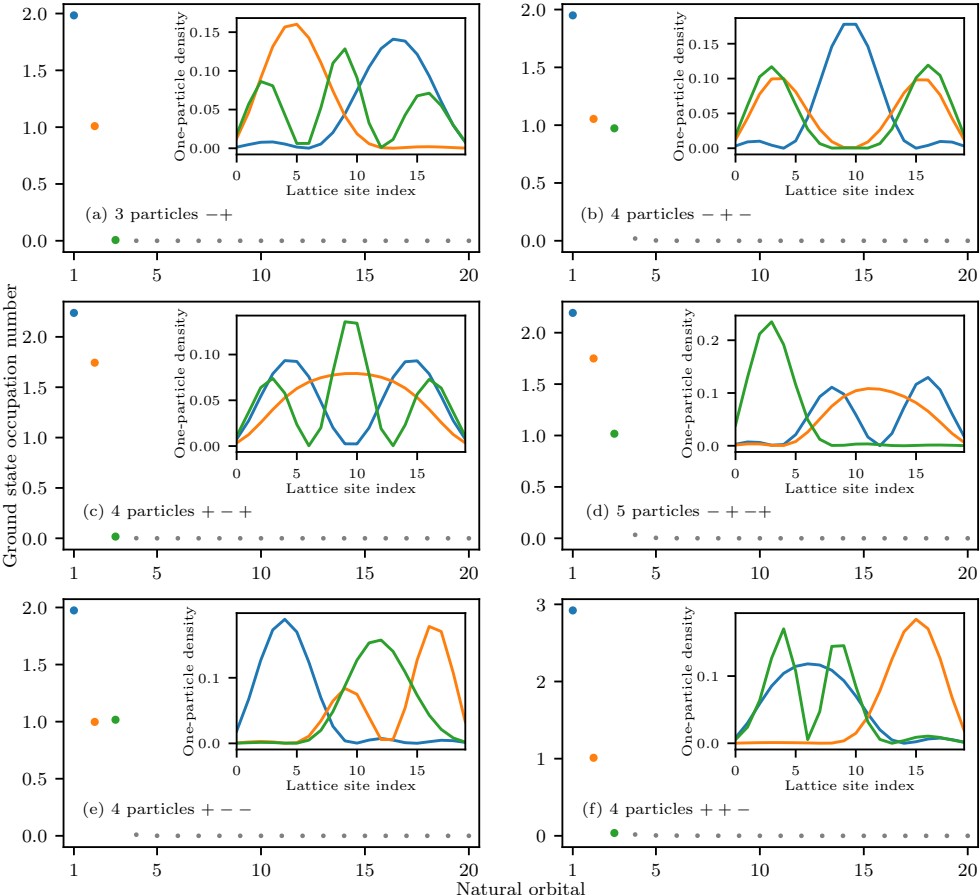

Figure 10: Ground state natural orbitals (insets) and their occupation numbers (main plots) for our traid anyon lattice model (14, 21) on 20 lattice sites without on-site interaction ($U = 0$), obtained from diagonalizing the single-particle density matrix $\langle t_i^\dagger t_j \rangle$ with the traid anyon operators defined in Eqs. 17 and 22.

the natural orbitals and their occupations, we find further evidence for the emergence of approximate fractional exclusion statistics and an explanation for the observed decrease of the chemical potential in every other step.

The emergence of semionic *exclusion* statistics can be understood intuitively as a consequence of the staggered *exchange* statistics of the traid anyons. Namely, when the rightmost particles form a 'bosonic' pair, a newly added particle comes with a minus sign and thus plays the role of a new single 'fermion' on the right edge. In turn, when the rightmost particle resembles a single 'fermion', the added particle comes with a plus sign, so that it can occupy the same space as the former rightmost particle (both form a new 'bosonic' pair).

It is straightforward to generalize these ideas to non-staggered representations. We then expect to find scenarios where 3 (or more) neighboring particles behave like bosons to each other, e.g. for ($++-++-...$). Also mixed fractional statistics can be imagined, e.g. for ($+-++-+-...$) pairs and triples of particles are expected to be in the same state. Such cases will be discussed in the next subsection.

## 4.3 Natural orbitals

The natural orbitals are defined as the eigenvectors of the single-particle density matrix (SPDM) $\langle c_i^\dagger c_j \rangle$, where $c_i^\dagger$ and $c_j$ are generic creation and annihilation operators, respectively.

The corresponding eigenvalues are the natural orbitals' occupation numbers that represent how many particles occupy each natural orbital on average. For the many-body ground state of free bosons ($c_i = b_i$) the single eigenvector with non-zero eigenvalue of the SPDM would be the state-space vector of the single-particle ground state and the corresponding eigenvalue would give the occupation number $N$ of that state. For free fermions, one finds $N$ eigenvectors with non-zero eigenvalues of 1, corresponding to the singly-occupied single-particle energy eigenstates forming the Fermi sea. Considering interacting bosonic or fermionic systems, the natural orbitals no longer correspond to the single-particle energy eigenstates and the occupation numbers of the natural orbitals are generally not integer-valued anymore. For free particles with fractional exclusion statistics, one would expect eigenvalues corresponding to occupations that take integer values larger than one, but smaller or equal than an allowed maximum occupation (e.g. two for so-called semions). Below, we indeed find approximately such behaviour, with small deviations that we attribute to the presence of interactions associated with the number-dependent tunneling even for $U = 0$.

In order to investigate the properties of the traid anyons, we need to consider the SPDM with respect to the traid-anyon operators,

$$\langle t_i^\dagger t_j \rangle = \langle b_i^\dagger e^{i\pi(M_i - M_j)} b_j \rangle \,. \tag{23}$$

Note that it is different from the bosonic one, $\langle t_i^\dagger t_j \rangle \neq \langle b_i^\dagger b_j \rangle$. An intuitive understanding of how the occupation numbers for hypothetical free traid anyons would look like for a given traid group representation $(\tau_1, \ldots, \tau_N - 1)$ can be gained as follows: Whenever there is a string of $\tau_i = +1$ of length $m$ in a representation, then those $m + 1$ particles exchange like bosons and can all occupy the same state. Whenever there is $\tau_i = -1$ in a representation, a new orbital must be added to account for the antisymmetric exchange. For example with $N = 3$, the alternating representation $(+-)$ would have one orbital localized on the left with occupation number 2 corresponding to the two particles with boson-like exchanges, and one orbital with occupation number 1 localized on the right side. The representation $(-+)$ would have the same occupation numbers as $(+-)$ but mirrored orbitals. For $N = 4$, the ground state of representation $(-+-)$ would have three orbitals, one with occupation number 2 localized in the center, and two with occupation number 1 localized on the sides, whereas $(+-+)$ would have two orbitals with occupation number 2 localized on either side. Generally, all alternating representations will have a maximum occupancy of 2, i.e., semionic exclusion statistics. The reasoning presented in this paragraph has to be taken with a grain of salt, as the actual natural orbitals are delocalized (as is known for instance for the case of free fermions). But it provides an intuitive understanding of the expected fractional exclusion statistics, as we indeed find it in our numerical results below (up to corrections which we attribute to the presence of interactions even for $U = 0$ associated with the number-dependent tunneling).

Our numerical results are depicted in Fig. 10. We display the occupation numbers of the natural orbitals (main plots), as well as the one-particle densities of the first three natural orbitals ordered by their occupation numbers (insets). While Figs. 10(a-d) show examples of traid anyons with alternating hopping signs, specified by Eqs. 14 and 17a, Figs. 10(e,f) additionally show two different nontrivial abelian representations for 4 particles, namely $(+--)$ and $(++-)$, defined by the generalized traid anyon lattice model (21, 22). In analogy to the ground state densities and energies discussed above, the densities of the natural orbitals corresponding to mirrored abelian representations [e.g., $(+-)$ and $(-+)$] are also mirrored with identical occupation numbers.

The most striking feature of the natural orbitals in the traid anyon picture is their (near) integer-valued occupation numbers that approximately conform to the free traid-anyon hypothesis described above. In particular, for the three-particle case shown in Fig. 10(a), we find occupation numbers of 1.984 and 1.009 for the first two natural orbitals. As we increase

$N$, in Fig 10(b)-(d), the expected pattern remains, although there are larger deviations from the predicted integer values. This can be interpreted as an (approximate) constraint of a maximum of two traid anyons occupying each natural orbital for the alternating representations. This behaviour provides further indication of an approximate emergent Haldane-like fractional semionic exclusion statistics in the traid-anyon-Hubbard model (14), as we discuss in Sec. 4.2 and App. D. We note that other (i.e. non-staggered) traid representations can also lead to a different generalized exclusion statistics, as exemplified in Fig. 10(e) and (f), where the occupation numbers again approximately conform to the integer values of the free traid anyon hypothesis.

The squared absolute value of the natural orbitals, as they are shown in the insets of Fig. 10, provide further insight into how the ground-state density distributions shown in Figs. 7 and 8 are formed. The shape of the natural orbitals also provides a possible explanation for the observation that the chemical potential in Fig. 9 decreases every other time when a particle is added to the system to form a 'bosonic' pair with a previously fermion-like single particle. Namely, in Fig. 10, one sees that the shape of the natural orbitals depends on the number of particles in the system. In particular, we find that the width of the natural orbitals tends to increase with their occupation, so that the kinetic energy is lowered for multiply-occupied orbitals relative to that of singly-occupied ones, suggesting a mechanism for the reduction of the chemical potential. This behaviour is clearly different from that found for the ground state of both free bosons and fermions, for which the natural orbitals are not number-dependent and simply given by the single-particle eigenstates.

If this broadening of multiply-occupied orbitals can be interpreted as a signature of some form of (effective) attractive interactions in the system, it might also explain the deviations from perfectly quantized occupations of the natural orbitals visible in Fig. 10. The origin of these interactions must be related to the form of the commutation relations of the traid operators (17). Different from the standard bosonic and fermionic ones, they involve 'interaction' terms, i.e. terms that involve products of more than two annihilation and creation operators. One immediate consequence is, that the commutation relations become basis-dependent. That is, transformations $t_\alpha = \sum_i \langle \alpha | i \rangle t_i$ to a basis of single particle states $|\alpha\rangle$ (for instance the basis of natural orbitals), other than the local Wannier basis $|i\rangle$ defining the lattice sites, change the form of the commutation relations.

## 5 Continuum limit

In section 3 we constructed a lattice model (14) using bosons with non-local, density-dependent Peierls phases that corresponds to abelian representations of the traid group. Here we derive the continuum limit of this model with staggered exchange phases (14). This approach can be easily generalized also to other abelian representations of the traid group. Similar to the Tonks-Girardeau model in which one-dimensional fermions can be mapped to hard-core bosons, we find that in the continuum limit abelian traid anyons can be expressed in terms of bosons with hard-core contact interactions. Importantly, these interactions depend on the ordinality of the particles and are, therefore, non-local.

We find that the eigenstates of the continuum model can be directly mapped to traid-anyon wave functions, as they were constructed previously from general considerations [16,17]. This mapping is similar to that from hard-core bosons to fermions. Namely, in our model, hard-core interactions occur whenever two particles meet and exchange like fermions. As a result, the many-body wave function becomes zero at this coincidence, allowing to flip the sign of the wave function, when mapping between bosons and traid anyons, without increasing the energy. The fact that we recover previous results from the continuum limit of our model system provides an a-posteriori justification of the construction of our model.

## 5.1 Continuum limit of the lattice model

We take the continuum limit of our lattice model by letting the lattice spacing go to zero,

$$d \to 0, \tag{24}$$

while keeping the physical length $l = Ld$ constant. We will also see that the effective mass of the continuum particles $m^*$ is defined by $Jd^2 = \hbar^2/(2m^*)$[3] and that the Hubbard interactions of strength $U$ give rise to a contact interaction potential $g\delta(x)$ of strength $g = Ud$. Thus, we have to scale $L$, $J$, and $U$ like

$$L = \frac{l}{d}, \qquad J = \frac{\hbar^2}{2m^*}\frac{1}{d^2}, \qquad U = \frac{g}{d}, \tag{25}$$

with $m^*$ and $g$ held constant. Moreover, we have to replace sums by integrals,

$$\sum_{j=1}^{L} \to \frac{1}{d}\int_0^l dx, \tag{26}$$

and bosonic annihilation and creation operators $b_j$ by bosonic field operators $\psi(x)$, according to [31]

$$\frac{b_j^{(\dagger)}}{\sqrt{d}} \to \psi^{(\dagger)}(x), \tag{27}$$

where $[\psi(x), \psi^\dagger(x')] = \delta(x - x')$.

It is straightforward to obtain the continuum limit of the on-site Hubbard interactions

$$H_{\text{os}} = \frac{U}{2}\sum_j b_j^\dagger b_j^\dagger b_j b_j \to \frac{g}{2}\int dx\, \psi^\dagger(x)\psi^\dagger(x)\psi(x)\psi(x) = \frac{g}{2}\int dx\, :\rho^2(x):, \tag{28}$$

where we have introduced the density operator $\rho(x) = \psi^\dagger(x)\psi(x)$ as well as the usual convention that $:\bullet:$ means normal ordering of all operators between the colons, i.e. all creation operators are positioned to the left of the annihilation operators.

Because of the number-dependent Peierls phases, the continuum limit of the tunneling term in Eq. (14) is more involved. Restricting ourselves to the case $I = 0$, it reads

$$H_{\text{tun}} = -J\sum_j^L \left( b_{j+1}^\dagger e^{i\pi N_j n_{j+1}} b_j + \text{h.c.} \right). \tag{29}$$

At the end, we will reintroduce the dependence on $I$, simply via the substitution $N_j \to N_j + I$. Calculating the continuum limit of the braid-anyon-Hubbard model, Bonkhoff et al. managed to preserve the crucial symmetry of the anyonic exchange phase under the transformation $\theta \to \theta + 2\pi$ in the continuum, by normal ordering the operators in the complex phase factor before taking the limit [15]. We closely follow their approach in deriving the continuum limit of the traid anyon lattice model. In our case, we want to preserve the symmetry of the complex phase factor under the transformation $N_i \to N_i + 2$. Thus, we calculate the continuum limit of the 'number-to-the-left' operator $N_i$ independently and then write all other operators in the number-dependent hopping phase in normal order before taking the limit. The continuum limit of this operator is given by

$$N_j = \sum_{k \leq j} b_k^\dagger b_k \to \int_0^x dx'\, \psi^\dagger(x')\psi(x') = \int_0^x dx'\, \rho(x') \equiv N(x), \tag{30}$$

---

[3]This result follows also directly from expanding the dispersion relation of free bosons with respect to the lattice spacing, $\varepsilon(k) = -2J\cos(dk) \simeq -2J + Jd^2k^2 \equiv \frac{\hbar^2 k^2}{2m^*} + \text{const.}$

where $x \equiv jd$. Analogous to the discrete case, the operator $N(x)$ counts the number of particles to the left of position $x$. In particular, one has $N(l) = N$.

Next, we expand the exponential phase factor:

$$H_{\text{tun}} = -J \sum_j b_{j+1}^{\dagger} \left( \sum_{q=0}^{\infty} \frac{(i\pi)^q}{q!} N_j^q n_{j+1}^q \right) b_j + \text{h.c.} \tag{31}$$

The powers of the number operators $n_{j+1}$ can now be written in the normal ordered form

$$n_{j+1}^q = \sum_{m=0}^{\infty} S(q,m) d^m \left( \frac{b_{j+1}^{\dagger}}{\sqrt{d}} \right)^m \left( \frac{b_{j+1}}{\sqrt{d}} \right)^m, \tag{32}$$

with the Stirling numbers of the second kind $S(q,m)$ [15, 32]. We now take the continuum limit, approximating the field operators at $x + d$ as:

$$\psi(x+d) \simeq \psi(x) + d\frac{\partial}{\partial x}\psi(x) + \frac{d^2}{2}\frac{\partial^2}{\partial x^2}\psi(x) \equiv \psi + d\psi_x + \frac{d^2}{2}\psi_{xx} \tag{33}$$

(note that higher order terms will not contribute in the continuum limit), and finally sort all resulting terms by their power in the lattice spacing $d$. The terms of zeroth order in $d$ are

$$-J \int dx \left( \psi^{\dagger}\psi + \text{h.c.} \right) = -2JN. \tag{34}$$

This amounts to a constant energy contribution, which we can neglect in the final continuum model.

For the first-order terms we find

$$-2Jd \int dx (\cos(\pi N(x)) - 1)\psi^{\dagger}\psi^{\dagger}\psi\psi = \frac{\hbar^2}{m^*} \int dx \frac{1 - \cos(\pi N(x))}{d} : \rho^2 : . \tag{35}$$

This term describes a diverging (i.e. hard-core) two-body contact interaction for even values of $N(x)$ due to the $1/d$ dependence.

Finally, the second-order terms read

$$-\frac{\hbar^2}{m^*} \int dx \left[ (1 - \cos(\pi N(x))) : \rho^3 : + \frac{1}{2} \left( \psi_{xx}^{\dagger}\psi + \psi^{\dagger}\psi_{xx} \right) \right], \tag{36}$$

corresponding to a kinetic energy term and a three-body interaction term. All other terms are of order $\mathcal{O}(d^3)$ and vanish in the continuum limit.

Summing up all terms (and including the continuum limit of the on-site interaction term in Eq. (14)), we obtain the final continuum Hamiltonian

$$H_{\text{cont.}} = \int dx \left[ -\frac{\hbar^2}{2m^*}\psi^{\dagger}\partial_x^2\psi + V_2 : \rho^2 : + V_3 : \rho^3 : \right], \tag{37}$$

with the two-body and three-body interaction coefficients

$$V_2(x) = \frac{\hbar^2}{m^*d} (1 - \cos[\pi(N(x) + I)]) + \frac{g}{2}, \tag{38}$$

$$V_3(x) = \frac{\hbar^2}{m^*} \left( \cos[\pi(N(x) + I)] - 1 \right), \tag{39}$$

where we reintroduced the index $I$, determining which of the two abelian representations of the traid group with alternating exchange phases the model realizes.

From the two-body interaction coefficient $V_2$ (38), we can see that two particles exchanging like fermions in the lattice model (i.e., where $N(x) + I$ is odd), now exhibit a hard-core interaction in the continuum limit. On the other hand, particles that exchange like bosons ($N(x) + I$ is even) do not interact at all with one another in the continuum model (if $g = 0$). The hard-core two-body interaction between alternating pairs of particles also implies a three-body hard-core constraint, which is characteristic of the traid group. Thus, it emerges naturally in the continuum limit of the traid anyon-Hubbard model (14), even if we do not impose it by additional potentials or constraints. As a result, in the $d \to 0$ limit, we can neglect the finite-strength three-body interaction term $V_3(x)$ in our alternating traid anyon model and simply write

$$H_{\text{cont.}} = \int dx \left[ -\frac{\hbar^2}{2m^*} \psi^\dagger \partial_x^2 \psi + V_2(x) : \rho^2 : \right]. \tag{40}$$

The emergence of the contact interactions with infinite (zero) strength for two particles that behave like fermions (bosons) to each other is also consistent with previous results of Refs. [12, 33]. There it was shown for the braid-anyon-Hubbard model, that the combination of number-dependent tunneling with anyonic phase $\theta$ and on-site Hubbard interactions $U$ gives rise to an effective contact interaction, whose strength diverges for $\theta \to \pi$ and vanishes when both $\theta = 0$ and $U = 0$.

Comparing the continuum limit (37) of the traid-anyon-Hubbard model to the continuum limit of the braid-anyon-Hubbard model [15], we see that we recover the same structure of the Hamiltonian, when we make the identification $\theta \to \pi(N(x) + I)$. The crucial difference between the two models is that in the case of the traid anyons the coefficients $V_2$ and $V_3$ are promoted to operators, resulting in non-local interactions between the particles. Moreover the current interaction term that arises in the continuum limit of the braid-anyon-Hubbard model vanishes for our model. Thus, in contrast to the continuum model of the braid-anyon-Hubbard model, the traid-anyon-Hubbard model constructed here gives rise to a Galilean invariant Hamiltonian.

A possible explanation for this difference is that the breaking of Galilean invariance via a dependence on the current operator originates from the requirement to distinguish exchange processes corresponding to the left particle moving over or under the right one, as depicted in Figs.2(b) and (c). While in the braid-anyon-Hubbard model these processes can be distinguished via different paths that form a non-contractable loop in the discrete configuration space [see Fig. 5], this is not possible in the continuum limit. Thus, the dependence on the current operator (also found in the chiral BF and Kundu models) can be interpreted as a way to "translate" the distinction between these two processes in the lattice where two particles exchange either via leftward or rightward tunneling to the continuum. A similar observation on the necessity for breaking Galilean invariance to implement fractional exchange statistics in 1D models has been made for excitons in the Haldane-Shastry model [34, 35]. However, for traid anyons this is not the case, suggesting that they correspond to a more natural (intrinsic) definition of anyons in the continuum in 1D, as is also suggested by the topology of the continuum strand diagrams discussed above.

## 5.2   Continuum model in first quantization

From (37), we can derive the bosonic continuum model in first quantization. First consider the Hamiltonian restricted to the sector $\mathcal{X}_{12\cdots N} \subset \mathbb{R}^N$ where the particle labels coincide with particle order $x_1 \leq x_2 \leq \cdots \leq x_N$. The Hamiltonian that realizes the alternating representations

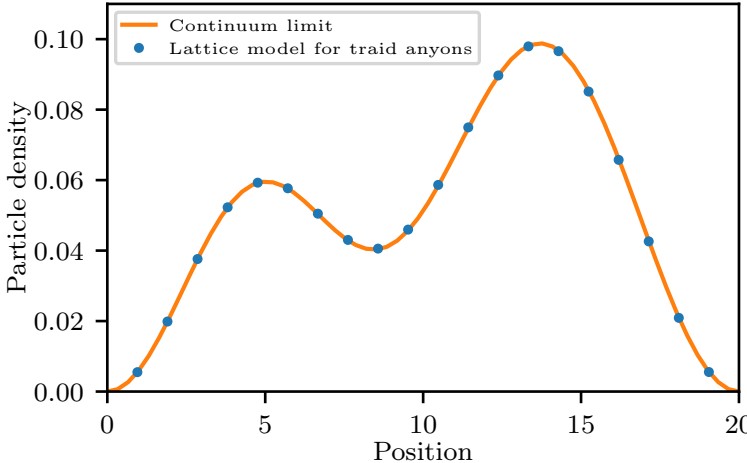

Figure 11: Comparison of the ground state densities of a system with 3 particles and traid group representation (−+) for both the traid anyon lattice model (14) and its continuum limit (41). The lattice system contains 20 sites. The continuum model was solved with a finite difference method. The lattice model densities were rescaled for comparison.

characterized by $I$ has the form

$$H_{\text{cont.}}^{\text{FQ}} = \sum_j \left( -\frac{\hbar^2}{2m^*} \frac{\partial^2}{\partial x_j^2} + \tilde{V}_2(j)\delta(x_j - x_{j+1}) \right),$$ (41a)

where the two-body interaction coefficient $\tilde{V}_2(j)$ is no longer an operator but an order-dependent number

$$\tilde{V}_2(j) = \frac{\hbar^2}{m^*d}(1 + \cos(\pi(j+I))) + \frac{g}{2}.$$ (41b)

The specific form chosen for the coefficient $\tilde{V}_2(j)$ is somewhat arbitrary; any function that alternates between 0 and 2 depending on the particle order index $j$ would work for the expression in parentheses. In another ordering sector $\mathcal{X}_{p_1 p_2 \cdots p_{N-1}}$, where the labelled particles are in the order $x_{p_1} \leq x_{p_2} \leq \cdots \leq x_{p_N}$, the only change to $H_{\text{cont.}}$ is that the delta function in Eq. 41a becomes $\delta(x_{p_j} - x_{p_{j+1}})$.

In the limit $d \to \infty$, the coefficient $\tilde{V}_2(j)$ forces a hard-core constraint between particles that have antisymmetric exchanges (i.e. that behave like fermions to each other). The alternating hard-core two-body interactions exclude triple coincidences and so again we choose to leave out the three-body interaction term from Eq. 37 in Eq. 41a. Note that the total Hamiltonian only depends on the relative coordinates and the interactions are invariant under Galilean transformations and permutations of particle labels. Time reversal invariance is immediate from the real form of Eq. 41, and spatial inversion leads to the same results as for the lattice model: Invariance for $N$ even and mapping between $I = 0$ and $I = 1$ for $N$ odd.

The ground state of this model can be easily calculated numerically, e.g. with a finite difference method. This allows us to compare the continuum model directly to the lattice model for traid anyons (14). We compute the ground state densities for both models and plot them in Fig. 11 (rescaling them for comparison). We see that the densities of both models match exactly, indicating that the continuum model (41) accurately captures the traid anyon characteristics in the low-energy, dilute limit that we observed in the lattice model.

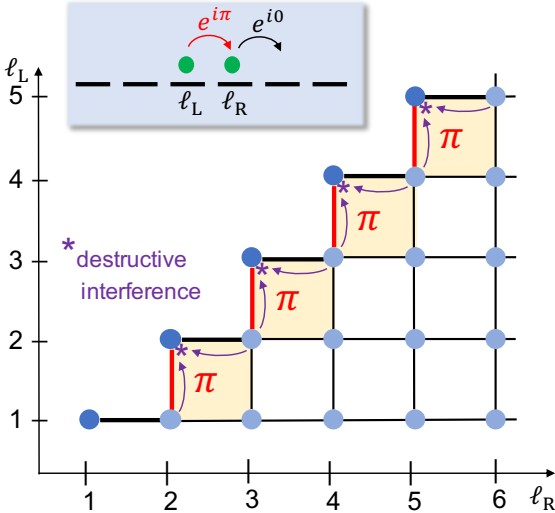

Figure 12: Dynamics of two identical particles on a lattice that behave like pseudo-fermions to each other. The positions of the left and right particle are labeled by $\ell_L$ and $\ell_R$, respectively, with $\ell_L \leq \ell_R$ (inset). The configuration space is plotted with dark (light) blue bullets corresponding to states, where both particles occupy the same (different) lattice site(s). Vertical (horizontal) lines represent tunneling processes of the left (right) particle. Thick lines correspond to tunneling matrix elements to or from a state with a doubly occupied site; these are enhanced by a factor of $\sqrt{2}$. Tunneling along the red lines is associated with an extra phase of $\pi$, resulting from number-dependent tunneling and giving rise to the pseudo-fermionic nature of the two particles. This leads to a flux of $\pi$ through the yellow-shaded plaquettes. In the continuum limit, the wave function can at most vary infinitesimally from lattice site to lattice site, because otherwise the kinetic energy would diverge. Thus, tunneling processes to a dark blue state with two particles at the same site will destructively interfere. In this way, two-particle coincidences are suppressed.

## 5.3 Origin of emergent hard-core interactions

The appearance of the delta-function-type contact interactions of infinite strength between two particles that behave like pseudofermions to each other can be understood intuitively from the lattice model. Let us consider a reduced model for two particles only, which locally captures the state space of the $N$-particle traid-anyon-Hubbard model close to a coincidence of two particles that are pseudofermions to each other. This situation is plotted in Fig. 12 (see caption for a description) and also captures the pseudofermion limit $\theta = \pi$ of the braid-anyon Hubbard model. The destructive interference prohibiting two particles to occupy the same site when approaching the continuum limit is associated with the appearance of plaquettes with $\pi$-flux at the edge of configuration space. These are indicated by the yellow shading.

In the continuum limit, the wave function can vary at most infinitesimally from site to site, because otherwise the kinetic energy would diverge. Therefore, tunneling processes that would create a doubly occupied site (dark blue bullets) are prohibited by destructive interference, as a result of the fact that tunneling matrix elements represented by red and black lines have opposite sign. In the continuum limit two-particle coincidences are, thus, suppressed completely. This effect is taken care of by the delta-function interaction of infinite strength. Note that for two particles that behave like bosons to each other, tunneling matrix elements described by red lines have the same sign as all the others, so that no plaquette flux is induced, this effect vanishes and no interactions appear in the continuum limit.

### 5.4 General abelian representations for traid anyons

The derivation of the continuum limit can be generalized to arbitrary abelian representations of the traid group. As discussed in section 3 for the traid anyon lattice model, other representations are realized by constructing more complicated polynomials of the operator $N_j$ (15) in the number-dependent hopping phase: $e^{i\pi F(N_j)n_{j+1}}$. Analogous to the case of alternating exchange phases, we can then preserve the symmetry under the transformation $F(N_j) \to F(N_j) + 2$ by normal ordering the other operators in the exponential before taking the limit. The resulting Hamiltonian has the same structure as Eq. 37, with the new interaction coefficients

$$V_2(x) = \frac{\hbar^2}{m^*d}\big(1 - \cos[\pi F(N(x))]\big) + \frac{g}{2}, \tag{42a}$$

$$V_3(x) = \frac{\hbar^2}{m^*}\big(\cos[\pi F(N(x))] - 1\big). \tag{42b}$$

The polynomial $F(N)$ is constructed in the lattice model so that it takes on even integer values if a pair of particles exchanges like bosons and odd integer values if a pair of particles exchanges like fermions. Therefore, analogous to the case of alternating exchange phases, bosonic pairs do not interact in the continuum limit, whereas pairs of particles behaving like fermions experience hard-core repulsion. Alternatively, the first quantization equivalent of (42a) for the general abelian traid representation $\tau = (\tau_1 \cdots \tau_{N-1})$ also could be expressed in the ordering sector $\mathcal{X}_{12\cdots N}$ as

$$\tilde{V}_2(j) = \frac{\hbar^2}{m^*d}(1 - \tau_j) + \frac{g}{2}. \tag{43}$$

For abelian representations of the traid group, with more than two phases $\tau_i = +1$ next to each other, i.e. $\tau_i = \tau_{i+1} = +1$ or $(\cdots + + \cdots)$, one finds sequences of three or more neighboring particles that behave like bosons to each other. In these cases, the continuum limit of our traid-anyon-Hubbard model does not possess emergent three-body hard-core interactions within these bosonic sequences, unless we include such a three-body hard-core constraint by hand. A proper continuum limit is approached even if we do not enforce three-body hard-core interactions on the level of the lattice model.

This raises the interesting question of whether the definition of traid anyons in the continuum actually requires three-body hard-core interactions between all particles, or whether they are required only among those triples of neighboring particles that do not all behave like bosons to each other. The continuum limit of the interaction coefficient $V_3(x)$ (42b), which is not hard-core, suggests that that the latter is true. An alternative approach is to include a three-body hard-core constraint on the level of the tight-binding Hamiltonian as has been done also here and previously also in Refs. [36, 37]. This leads to a continuum limit with a three-body hard-core constraint among all triplets that is consistent with the traid-anyon wave functions constructed previously [16, 17]. The differences between these approaches is a subject for future investigation.

### 5.5 Mapping between bosonic and traid anyon wave functions

In this last subsection on the continuum limit, we discuss the mapping of bosonic states that evolve according to the Hamiltonian (41) and anyonic states that obey the symmetries given by the traid group [16, 17]. For simplicity, we will consider the case of three particles with $g = 0$ as the simplest non-trivial scenario. As shown in Fig. 13, the structure of interactions is

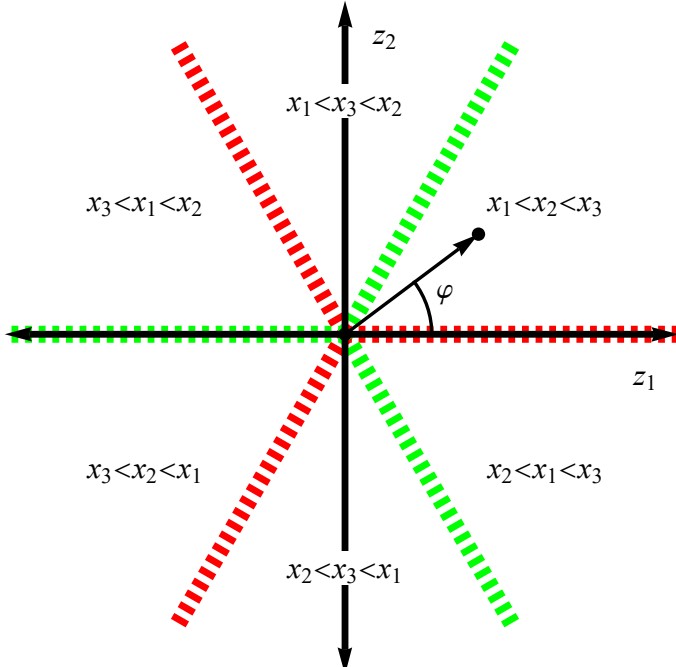

Figure 13: The relative configuration space for three particles in a coordinate system (44). The configuration space is divided into six sectors depending on the order of the particles, indicated by inequalities of the form $x_{p_1} < x_{p_2} < x_{p_3}$. Two-body coincidences $x_i - x_j = 0$ occur at the colored, dashed lines separating the sectors at $\varphi \in \{-2\pi/3, -\pi/3, 0, \pi/3, 2\pi/3\}$, where $\tan\varphi = z_2/z_1$. The dashed red half-lines at $\varphi = 0$, $2\pi/3$, and $-2\pi/3$ correspond to configurations where the leftmost and center particles coincide. The green half-lines at $\varphi = \pi/3$, $\pi$, and $-2\pi/3$ correspond to configurations where the center and rightmost particles coincide. A hard-core three-body constraint excludes the origin of this coordinate system.

revealed by transforming from the particle coordinates $\mathbf{x} = (x_1, x_2, x_3) \in \mathbb{R}^3$ to Jacobi coordinates $\mathbf{z} = (z_0, z_1, z_2)$:

$$
\begin{aligned}
z_0 &= \frac{1}{\sqrt{3}}(x_1 + x_2 + x_3), \\
z_1 &= \frac{1}{\sqrt{6}}(2x_3 - x_1 - x_2), \\
z_2 &= \frac{1}{\sqrt{2}}(x_2 - x_1),
\end{aligned}
\tag{44}
$$

where $z_0$ is proportional to the center-of-mass, $z_2$ proportional to the relative position of the particles labelled 1 and 2, and $z_1$ proportional to the relative position of particle 3 with the center-of-mass of particles 1 and 2, and all of these have been normalized so that the transformation $\mathbf{x} \to \mathbf{z}$ is orthogonal and $\sum_{j=1}^{3} \partial^2/\partial x_j^2 = \sum_{j=0}^{2} \partial^2/\partial z_j^2$. Converting to polar coordinates in the relative plane,

$$
\tan\varphi = \frac{z_2}{z_1}, \rho = \sqrt{z_1^2 + z_2^2},
\tag{45}
$$

gives the relative radius $\rho$, describing the size of the three-body configuration, and the angular coordinate $\varphi$ describing the relative shape [38].

The relative coordinate plane is depicted in Fig. 13 and the symmetry revealed there explains why Jacobi coordinates are convenient. As $\varphi$ is varied from $-\pi$ to $\pi$ for fixed $\rho$, the three particles execute a cyclic exchange through six two-body coincidences. Three angles

denoted $\alpha_1$ correspond to coincidences of the first two particles ($x_i = x_j < x_k$),

$$\alpha_1 \in \left\{ -\frac{2\pi}{3}, 0, \frac{2\pi}{3} \right\}, \tag{46a}$$

and three angles denoted $\alpha_2$ correspond to coincidences of the second two particles ($x_k < x_i = x_j$),

$$\alpha_2 \in \left\{ -\frac{\pi}{3}, \frac{\pi}{3}, \pi \right\}, \tag{46b}$$

with $i \neq j \neq k$. Expressed in Jacobi cylindrical coordinates, any bosonic wave function $\Psi(z_0, \rho, \varphi)$ must satisfy the following boundary conditions at both $\alpha_1$ and $\alpha_2$ to be symmetric under any two-particle exchange:

$$\lim_{\epsilon \to 0} \Psi(z_0, \rho, \alpha_i + \epsilon) = \lim_{\epsilon \to 0} \Psi(z_0, \rho, \alpha_i - \epsilon), \tag{47a}$$

$$\lim_{\epsilon \to 0} \Psi'(z_0, \rho, \alpha_i + \epsilon) = -\lim_{\epsilon \to 0} \Psi'(z_0, \rho, \alpha_i - \epsilon), \tag{47b}$$

where $\Psi'(z_0, \rho, \varphi) = \partial \Psi / \partial \varphi$ and the extra minus sign in Eq. 47b accounts for the reflection of $\varphi$ in the derivative.

In the Jacobi relative coordinates, the Hamiltonian (41) for the sector $x_1 \leq x_2 \leq x_3$ has the form

$$H_{\text{cont.}} = -\sum_{j=0}^{2} \frac{\hbar^2}{2m^*} \frac{\partial^2}{\partial z_j^2} + (1 - \tau_1) \frac{\hbar^2}{\sqrt{2}m^*d} \frac{\delta(\varphi)}{\rho} + (1 - \tau_2) \frac{\hbar^2}{\sqrt{2}m^*d} \frac{\delta(\varphi - \pi/3)}{\rho}, \tag{48}$$

and symmetric forms in other sectors.[4] For traid representations with $\tau_1 = -1$, the red dashed half-lines at $\alpha_1$ in Fig. 13 represent infinite-strength, contact interactions between the first and second particle in the continuum limit $d \to 0$. This forces an additional condition on the wave function

$$\Psi(z_0, \rho, \alpha_1) = 0, \quad \text{for} \quad \tau_1 = -1. \tag{49a}$$

Similarly, for traid representations with $\tau_2 = -1$, the green half-lines at $\alpha_2$ represent infinite-strength delta functions between the second and third particle and this forces the condition

$$\Psi(z_0, \rho, \alpha_2) = 0, \quad \text{for} \quad \tau_2 = -1. \tag{49b}$$

Because the many-body wave function vanishes at certain coincidences, it costs no extra energy to flip the sign of the wave function on one side of such a coincidence. This allows us to construct maps from bosonic wave functions of Eq. 48 to fermionic or traid-anyonic wave functions $\tilde{\Psi}(z_0, \rho, \varphi)$ that are antisymmetric with respect to the exchange of the two leftmost particles, if $\tau_1 = -1$, and/or the two rightmost particles, if $\tau_2 = -1$:

$$\lim_{\epsilon \to 0} \tilde{\Psi}(z_0, \rho, \alpha_i + \epsilon) = \tau_i \lim_{\epsilon \to 0} \tilde{\Psi}(z_0, \rho, \alpha_i - \epsilon), \tag{50a}$$

$$\lim_{\epsilon \to 0} \tilde{\Psi}'(z_0, \rho, \alpha_i + \epsilon) = -\tau_i \lim_{\epsilon \to 0} \tilde{\Psi}'(z_0, \rho, \alpha_i - \epsilon). \tag{50b}$$

In the fermionic case ($--$), the analogy to the Tonks-Girardeau gas is evident; infinite-strength two-body interactions mimic the exclusion provided by fermion statistics. In contrast, for the anyonic representations ($+-$) and ($-+$), the two-body hard-core interactions depend on the relative location of the third particle. We will see below that, as a result, the mapping from bosonic to traid-anyon wave functions is not one-to-one.

---

[4]Note that in the intrinsic approach to topological exchange statistics (see App. C), indistinguishability effectively restricts the configuration space to only one of the six sectors depicted in Fig. 13.

To make this mapping more explicit, consider the special case when there is either no trapping potential or a harmonic trapping potential and there are no additional two-body interactions. Then, we can find exact solutions by recognizing that in the $d \to 0$ limit the Hamiltonian (48) separates in $z_0$, $\rho$, and $\varphi$ coordinates. Call $F(\varphi)$ the angular factor of the three-body wave function, i.e. $\Psi(z_0, \rho, \varphi) = \chi(z_0, \rho)F(\varphi)$. In between coincidences, there is no angular potential, so the angular wave function has the free form $F(\varphi) = A\exp(i\mu\varphi) + B\exp(-i\mu\varphi)$ for $\mu \geq 0$. Combining the bosonic requirement (47) with the representation-dependent two-body coincidence conditions (49), the lowest energy, unnormalized solutions $F_{(\tau_1, \tau_2)}(\varphi)$ for the representations $(\tau_1, \tau_2)$ are

$$F_{(++)}(\varphi) = 1, \tag{51a}$$

$$F_{(--)}(\varphi) = |\sin(3\varphi)|, \tag{51b}$$

$$F_{(+-)}(\varphi) = |\cos(3\varphi/2)|, \tag{51c}$$

$$F_{(-+)}(\varphi) = |\sin(3\varphi/2)|, \tag{51d}$$

corresponding to bosons $[(+, +)]$, fermions $[(-, -)]$, and traid anyons $[(+, -), (-, +)]$. The last three wave functions (51b-51d) are depicted by the solid lines in Fig. 14.

For the representation $(--)$, the antisymmetric boson-fermion map of Girardeau [39–41] maps the bosonic wave function (51b) depicted in Fig. 14(a) to the corresponding fermionic wave function by flipping the sign at each two-body coincidence. The resulting wave function $\tilde{F}_{(--)}(\varphi) = \sin(3\varphi)$, depicted by a dashed line in Fig. 14(a), has the symmetries under exchange of particle labels that we expect for fermions: $\tilde{F}(\alpha_i + \epsilon) = -\tilde{F}(\alpha_i - \epsilon)$ and $\tilde{F}'(\alpha_i + \epsilon) = \tilde{F}'(\alpha_i - \epsilon)$ for $\epsilon \to 0$. Note that it is possible to satisfy these requirements with single-valued functions on the interval $\varphi \in (-\pi, \pi]$.

In contrast, when we attempt to construct a similar map from the bosonic to the anyonic wave functions for the $(+-)$ and $(-+)$ representations, we find the function that satisfies (50) must be double-valued. For these representations, traid anyon exchange statistics impose the symmetries at $\alpha_1$ and $\alpha_2$: $\tilde{F}(\alpha_i + \epsilon) = \tau_i \tilde{F}(\alpha_i - \epsilon)$ and $\tilde{F}'(\alpha_i + \epsilon) = -\tau_i \tilde{F}'(\alpha_i - \epsilon)$ for $\epsilon \to 0$. Unlike the fermionic case, flipping the sign of the wave functions at the hard-core two-body coincidences no longer yields a single-valued map; the sign of the resulting function depends on where we begin the sequence of sign-flipping exchanges. For example, consider the function $F_{(+-)}$ (51c). One possibility for flipping the signs is depicted by the dashed line in Fig. 14(b), where $F_{(+-)}$ has been sign-flipped at the two-body hard-core coincidences at $\varphi = -\pi/3$ and $\pi/3$, giving the function $\tilde{F}_{(+-)}(\varphi) = \cos(3\varphi/2)$. However, choosing to start in the domain $\varphi \in [-\pi, -\pi/3]$ and antisymmetrizing across $\varphi = -\pi$ and $-\pi/3$ gives the opposite sign $\tilde{F}_{(+-)}(\varphi) = -\cos(3\varphi/2)$. This indicates that to implement the correct exchange statistics, the traid anyon wave function for the $(+-)$ representation must be a double-valued functions on the interval $\varphi \in (-\pi, \pi]$.

Note that the three-body wave functions $\tilde{F}(\varphi)$ that we have obtained here by taking the continuum limit of our lattice model, directly correspond to the continuum traid-anyon wave functions constructed previously in Ref. [16] from general properties of the traid group. This provides another justification of the intuitive construction of the lattice model presented here.

In the presence of non-quadratic trapping potentials or additional interactions between the particles, the above reasoning remains valid, except that separability is lost and the wave functions do not have a simple form like (51). We can also generalize Eqs. (47-50) to the case of more than three particles by requiring corresponding symmetries dictated by $\tau_i$ for each of the relative coordinates of neighboring particles, while keeping all other coordinates fixed. This mapping between bosons and traid anyons provides traid-anyon wave functions in the continuum and corresponds to the generalized Jordan-Wigner transformation that we employed for the traid-anyon-Hubbard model.

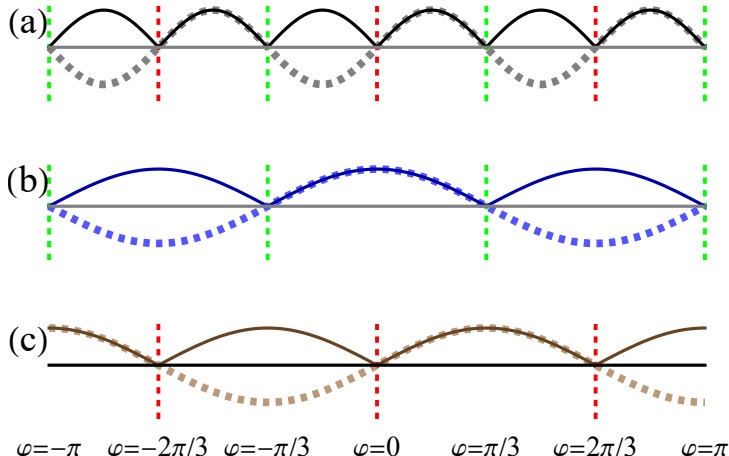

$\varphi=-\pi \quad \varphi=-2\pi/3 \quad \varphi=-\pi/3 \quad \varphi=0 \quad \varphi=\pi/3 \quad \varphi=2\pi/3 \quad \varphi=\pi$

Figure 14: This figure depicts the angular factor $F_{(\tau_1 \tau_2)}(\varphi)$ of bosonic three-body wave functions solving the Hamiltonian (48) for the abelian traid group representations $(--)$, $(+-)$, and $(-+)$, respectively. The solid lines correspond to the lowest energy solutions. The thicker dotted lines show corresponding traid-anyon wave functions $\tilde{F}_{(\tau_1 \tau_2)}(\varphi)$. In the pseudofermion case depicted in subfigure (a), infinite-strength contact interactions force nodes at every two-body coincidence for the black, solid wave function. The gray, thicker dotted line represents the angular wave function of the lowest energy free fermion state. This function has the same energy as its bosonic counterpart and is single-valued on the interval $\varphi \in (-\pi, \pi]$. It can be related to the bosonic wave function by the totally antisymmetric boson-fermion map of Girardeau [39]. The solid lines in subfigures (b) and (c) depict the bosonic wave functions corresponding to traid anyon representations $(+-)$ and $(-+)$, respectively. These have nodes at alternating two-body coincidences. The dotted lines represent one sheet of the corresponding multi-valued anyonic wave functions.

## 6 Conclusions

The main result of our paper is the construction of a lattice model for abelian traid anyons in one dimension. The starting point of this construction is the observation that the discreteness of the lattice's configuration space (which is given by Fock states with sharp site occupation numbers) allows to associate the exchange of two particles on neighboring sites with different paths in that space. The exchange can either be accomplished by the left particle hopping to the right onto the coincidence followed by a second hop to the right or by the right particle hopping to the left onto the coincidence followed by a second hop to the left. One can argue that distinguishing both processes, and associating them with the left or the right particle passing "in front" of the other one, allows to define the braiding of particles in one dimension and that this physics is realized by the (braid) anyon Hubbard model [11]. The phase picked up by the wave function under such particle exchange processes directly corresponds to a geometric phase around (or flux through) closed loops in configuration space. Abelian representations for braid anyons are characterized by a single angle $\theta$ that determines the exchange phase and smoothly interpolates between bosons ($\theta = 0$) and fermions ($\theta = \pi$). For abelian traid anyons, in turn, the phase $\theta_i$ associated with the exchange of the $i$th and $(i+1)$st particle from the left depends on the ordinality $i$, but can take two possible values, 0 or $\pi$ only, giving rise to a countable number of abelian representations $(\tau_1, \tau_2, \ldots, \tau_{N-1})$ with $\tau_i = \exp(i\theta_i)$.

In order to construct a lattice model exhibiting these traid anyon statistics, we first consider bosonic particles with non-local occupation-number-dependent hopping (or Peierls) phases.

These are chosen to give rise to the geometric phases in Fock space corresponding to the desired abelian representation of traid anyons. This approach resembles the composite-particle description of braid anyons in two dimensions, where anyons are mapped to charged bosons (or fermions) to which a flux tube is attached that gives rise to the geometric phase $\theta$ when one particle is moved around another one. In a second step, we then introduce traid-anyon annihilation and creation operators, with respect to which the kinetic term of the Hamiltonian becomes quadratic and local, and derive their commutation relations. This construction provides an intuitive approach to traid anyons.

Investigating the ground-state properties of this traid-anyon-Hubbard model, we find various indications also of an approximate Haldane-type exclusion statistics. For instance, we observe generalized Friedel oscillations, nearly integer occupations of the natural orbits, and a step-wise change of the chemical potential with particle number.

Finally, we take the continuum limit of our lattice model and find a Hamiltonian, whose eigenstates indeed correspond to traid-anyon wave functions, as they were constructed previously from the properties of the traid group [16]. This justifies *a posteriori* the construction of our model. Moreover, the continuum limit of the traid-anyon Hubbard model is found to be Galilean invariant. This is in clear contrast to the continuum limit of the braid-anyon-Hubbard model, which breaks Galilean invariance by a Hamiltonian that, like the Kundu model [42–47] and chiral BF model [48–52], depends on the current-density operator [15]. We attribute this difference to the fact that traid anyons are naturally defined also in the continuum in 1D using the intrinsic approach to topological exchange statistics. For braid anyons, on the other hand, this is not the case as they require to distinguish exchange processes with the left particle passing over or under the right one during an exchange, which in a 1D continuum system can only be mimicked dynamically by a velocity dependence.

Our results indicate that the possibilities for non-standard statistics might in some sense be even richer in one dimension than in two dimensions, at least when considering discrete lattice models. We have shown how 1D bosonic lattice models with density-dependent Peierls phases can implement abelian anyons that possess two completely different forms of exchange statistics, each corresponding to a different way of breaking the symmetric group: Braid statistics and traid statistics. In both cases, the theoretical link between the interacting bosonic lattice Hamiltonian and the 'free' anyonic Hamiltonian is provided by a Jordan-Wigner-like gauge transformation that can be expressed in the form

$$a_j = e^{i\Theta(N_j)}b_j, \tag{52}$$

where $\Theta(N_j)$ is a so-called non-local string operator that depends on $N_j$, the number of particles to the left of (and including) site $j$. For the braid-anyon Hubbard model, the function has the simple form $\Theta(N_j) = \theta N_j$ and the bosonic Hamiltonian possesses local interactions but has a continuum limit that is not Galilean-invariant and may not lead to a consistent boson-anyon mapping [41]. For the traid-anyon Hubbard model, the function has a more complicated form $\Theta(N_j) = -\pi f(N_j)$ where $f(N_j)$ is an integer function that depends on the traid group representation. The resulting bosonic Hamiltonian possesses non-local interactions that depend on $N_j$. It has a Galilean-invariant continuum limit and a boson-anyon mapping that is multivalued but well-defined. The precise form of $f(N_j)$ depends on whether we allow or exclude three-body (or more) coincidences on the lattice. Since the specific choice of $f(N_j)$ results in differing continuum limits, the role of the three-body constraints merits further investigation.

These two choices for $\Theta(N_j)$ were inspired by analogy with anyons in the continuum, and there are certainly other possible functions to consider. Whether these alternate forms would have physically-meaningful exchange statistics, locality properties, and continuum limits is an interesting question. One can also imagine more general string operators that depend not just on the total number of particles to the left $N_j$, but on their specific occupation-number configuration. However, as noted before, in the continuum limit the ordinality operator $N_j \to N(x)$

has a special topological property: It is invariant under homeomorphisms of a line. More generally, even indistinguishable particles can be given 'natural' labels on a line based on their ordinality, unlike particles in higher dimensions. While the notion of ordinality on a line may seem trivial, its topological origin may explain why 1D quantum models support such a wealth of integrable and statistical structures.

The chiral BF model as well as the braid-anyon Hubbard model have recently been realized experimentally with ultracold atoms [14, 52]. The construction of the traid-anyon Hubbard model presented here constitutes an important step also towards a first experimental realization of traid anyons in the laboratory. It is an interesting task for future research to design experimentally accessible schemes for the implementation of this (or related) traid-anyon models, though such an implementation will have to address the challenge of effectively realizing the required non-local interactions. Another interesting open question concerns the possibility of finding interacting one-dimensional systems, whose excitations behave like traid anyons.

## Acknowledgments

We thank Martin Bonkhoff and Philip Johnson for useful discussions.

**Funding information**  This research was funded by the Deutsche Forschungsgemeinschaft (DFG) via the Research Unit FOR 2414 under project No. 277974659. Moreover, N.H. acknowledges the support from the National Science Foundation under Grant No. NSF PHY-1748958 and the Deutscher Akademischer Austauschdienst, B.W. from the ERC grant LATIS and the EOS project CHEQS, and S.N. from the Provincia Autonoma di Trento, Q@TN, and the BMBF through the project 'MAGICApp'.

## A  Topological exchange statistics

Topological exchange statistics is an alternate approach to the Symmetrization Postulate for treating indistinguishable particles in first quantization. Instead of being imposed at the outset, the transformation properties of wave functions under particle exchanges are derived from the topology of configuration space. Topological exchange statistics can reproduce the results of the Symmetrization Postulate, but can also provide additional anyonic solutions in low-dimensional systems with hard-core interactions.

The key observation that underlies this approach to exchange statistics is that, when the classical configuration space $\mathcal{Q}$ for a physical system is not simply-connected, i.e. not all loops can be continuously deformed to a point, then in addition to single-valued wave functions on $\mathcal{Q}$, multi-valued wave functions may be necessary to describe the full Hilbert space of functions of $\mathcal{Q}$. These multi-valued functions can be formulated in several different frameworks [53–57].

The topological approach to exchange statistics was pioneered by researchers in path integration methods [53,58,59]. In 1977, Leinaas and Myrheim [1] first recognized the possibility for novel exchange statistics in low-dimensional systems; Goldin et al. independently rediscovered this result a few years later using a related approach based on current algebras [2]. The topological approach to exchange statistics was extended to orbifold configuration spaces in [16,17].

The topological approach is also called the *intrinsic approach* to exchange statistics because it does not introduce arbitrary particle labels for indistinguishable particles [60]. Whereas the Symmetrization Postulate quantizes the distinguishable particle configuration space $\mathcal{X} \subseteq \mathcal{M}^N$, the intrinsic approach quantizes the quotient space $\mathcal{Q} = \mathcal{X}/S_N$. In the quotient map from $\mathcal{X}$ to

$\mathcal{Q}$, points of $\mathcal{X}$ that differ by a permutation of labels are collapsed to same point in $\mathcal{Q}$. Unless $\mathcal{X}$ excludes all two-body coincidences, the quotient space $\mathcal{Q}$ is not naturally a manifold and is better described as an orbifold [16, 17, 60–63]. An advantage of the intrinsic approach is that, unlike a symmetry, true indistinguishability cannot be broken; the intrinsic approach treats particle labels as a gauge structure imposed on indistinguishable particles, not as a symmetry of configuration space [25, 64].

In the intrinsic approach, particle exchanges are concrete continuous trajectories interchanging the particles, not abstract label permutations. These exchange trajectories start and end at configurations that differ at most by particle labels and so, in the indistinguishable configuration space $\mathcal{Q}$, they form based loops. Under continuous deformation, equivalence classes of based loops form the fundamental (or first homotopy) group $\pi_1$ on a manifold; for orbifolds, such as $\mathcal{Q}$, this can be generalized to the orbifold fundamental group $\pi_1^*$ [65, 66]. Unitary irreducible representations of this group classify different quantizations of the configuration space, or in the case of particle exchanges, different superselection sectors.

When $\mathcal{M}$ is simply-connected (i.e., $\pi_1(\mathcal{M})$ is trivial) and there are no hard-core interactions, then $\pi_1^*(\mathcal{Q}) = S_N$ for any $\dim\mathcal{M}$ and the group of possible exchanges is equivalent to the group of particle permutations. In this case, the Symmetrization Postulate holds and there are only bosons and fermions with single-valued wave functions on $\mathcal{X}$ and standard exchange statistics.

## B Braid group and fractional exchange statistics

In 2D, hard-core two-body interactions create topological defects that make the configuration space not simply-connected. The paths of particle exchanges can wind around these defects, meaning that multiple *topologically inequivalent* exchanges may lead to the same permutation. The configuration space for $N$ indistinguishable particles on a plane $\mathbb{R}^2$ is

$$\mathcal{Q}_B \equiv \frac{\mathbb{R}^{2N} - \Delta_2}{S_N}, \tag{B.1}$$

where $\Delta_2$ is the set of points in $\mathcal{M}^N = \mathbb{R}^{2N}$ where at least two coordinates coincide. The space $\mathcal{Q}_B$ is a manifold; all coincidences $\Delta_2$ have been removed and so $\mathcal{Q}_B$ contains no singular orbifold points. In the simplest case of $N = 2$, the intrinsic configuration space $\mathcal{Q}_B$ is the direct product of a plane $\mathbb{R}^2$ for the center-of-mass coordinates and a cone with opening angle $\pi/3$ with an excluded tip for the relative coordinates [1].

The resulting fundamental group for $N$ particles is the braid group $\pi_1(\mathcal{Q}_B) = B_N$ [4]. As described in the main text, fractional exchange statistics are given by the abelian representations of the braid group (5) with phase $b_i \to \beta_i = \exp(i\theta_i)$. This exchange phase $\theta$ can be associated to the Aharonov-Bohm phase in the composite charged particle/flux tube model of Wilczek [67] that gave anyons their name. Fractional exchange statistics emerge for quasiparticle excitations in certain quantum liquid and quantum spin liquid models like the fractional quantum Hall effect [8] or the Haldane-Shastry spin-chain model [34, 35]. In addition to motivating the braid-anyon-Hubbard model [11], implementing fractional exchange statistics in one-dimensional systems provided the original motivation for several continuum models, including the recently experimentally-realized chiral BF model [48–52], the closely-related Kundu model (also called Lieb-Liniger anyons) [42–47], the Calogero-Sutherland anyon model [25, 68, 69], and hard-core anyon models [70–72]. Note that we have only discussed abelian braid anyons; non-abelian representations of $B_N$ are carried by multi-component, multi-valued wave functions, c.f. [73, 74].

In some of this literature on fractional exchange statistics, there exists an alternate definition to (5) for fractional exchange statistics where the transformation law for wave functions $\Psi(\mathbf{x})$ is defined on $\mathcal{X}$ instead of $\mathcal{Q}$. In the simplest case for two particles, this alternate choice for the transformation has the form $\Psi(x_1, x_2) = \exp(i\theta)\Psi(x_2, x_1)$, and this generalizes to more particles (c.f. [42, 72]). Iterating this form of the transformation again, we find $\Psi(x_1, x_2) = \exp(2i\theta)\Psi(x_1, x_2)$, showing this wave function is indeed multi-valued as one expects for an anyonic wave function. However, there is not a mechanism for implementing the inverse transformation (i.e, an 'under' path with phase $\exp(-i\theta)$ to match the 'over' path). Effectively, this equates both $b_i$ and $b_i^{-1}$ with the same permutation on $\mathcal{X}$, so this alternate transformation rule does not describe abelian representations of the braid group.

Anyonic wave functions $\Psi(\mathbf{q})$ can be lifted to functions on $\mathcal{X}$, but they will be multi-valued except for the special cases of fermions $\theta = \pi$ and bosons $\theta = 0$. Alternatively, one can work with single-valued functions on $\mathcal{X}$ for arbitrary $\theta$ at the cost of a gauge transformation. Famously, anyons in two dimensions with fractional exchange statistics can be transmuted into bosons or fermions with single-valued wave functions by introducing a magnetic gauge potential [6, 8].

## C  Traid group and abelian traid anyon exchange statistics

In analogy to the braid group, hard-core three-body interactions in 1D also introduce topological defects to configuration space that lead to a generalization of the symmetric group. The relevant intrinsic configuration space is

$$\mathcal{Q}_T = \frac{\mathbb{R}^N - \Delta_3}{S_N}, \tag{C.1}$$

where $\Delta_3$ is the locus of points in $\mathcal{M}^N = \mathbb{R}^N$ where at least three particle coordinates coincide. The orbifold fundamental group $\pi_1^*(\mathcal{Q}_T) = T_N$ for $N \geq 3$ indistinguishable particles is the traid group. Note that $\mathcal{Q}_T$ is an orbifold, not a manifold, because not all orbifold points in $\Delta_2$ have been excluded [16, 17].

The configuration space $\mathcal{Q}_T$ for the simplest case of $N = 3$ is represented in Fig. 15. In this depiction, it looks like a wedge $q_1 \leq q_2 \leq q_3$ with $q_i \in \mathbb{R}$ with an opening angle of $\pi/3$. Note that this picture is misleading at the orbifold points, as their non-trivial topology cannot be represented in this embedding. The two half-plane faces of the wedge correspond to configurations where the leftmost and rightmost pairs of particles coincide. The line of intersection of these half-planes is the excluded locus $\Delta_3$.

For the case of $N = 3$, the orbifold fundamental group is the traid group $T_3$ generated by two pairwise exchanges $t_1$ and $t_2$. In Fig. 15, these look like paths that start at a base point in $\mathcal{Q}_T$ and touch the orbifold singularities at $q_1 = q_2$ and $q_2 = q_3$, respectively. Because $t_i^2 = 1$, a loop that touches the same orbifold singularity twice can be contracted to a loop that does not touch the boundary. Therefore, all loops are either trivial, corresponding to the identity element of $T_3$, or are alternating sequences of $t_1$ and $t_2$. Aficionados of discrete groups will recognize that $T_3$ is isomorphic to the infinite dihedral group $D_\infty$, which has four one-dimensional irreducible representations corresponding to the four we have already described $(++)$, $(+-)$, $(-+)$, and $(--)$; c.f. Table. 1. There is also a 1-parameter family of two-dimensional irreducible representations characterized by the angle $\alpha$ between the two 'reflections' $t_1$ and $t_2$; these can be parameterized as

$$t_1 \rightarrow \begin{pmatrix} 1 & 0 \\ 0 & -1 \end{pmatrix}, \qquad t_2 \rightarrow \begin{pmatrix} \cos\alpha & \sin\alpha \\ \sin\alpha & -\cos\alpha \end{pmatrix}. \tag{C.2}$$

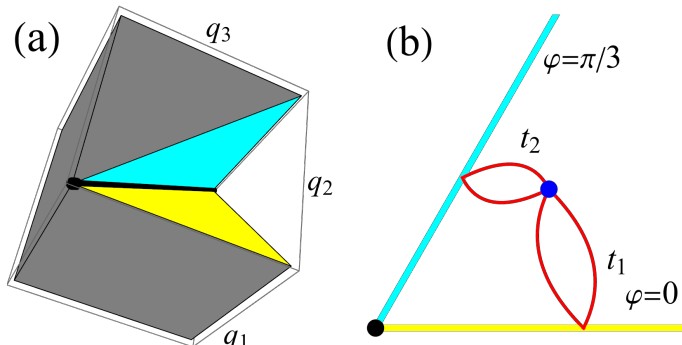

Figure 15: Subfigure (a) shows the configuration space for three indistinguishable particles on a line $\mathcal{Q}_T = (\mathbb{R}^2 - \Delta_3)/S_3$. See text for description; the coordinates for the sector are $q_1 \leq q_2 \leq q_3$ where $q_i$ is the $i$th particle from the left. The yellow boundary half-plane at the bottom is the orbifold locus $q_1 = q_2$; the cyan boundary half-plane at the top is the orbifold locus $q_2 = q_3$. The black diagonal is the three-body coincidence locus $\Delta_3$. Subfigure (b) shows the *relative* configuration space for $\mathcal{Q}_T$ along with two representative exchange paths for loops $t_1$ that touch the orbifold locus $q_1 = q_2$ and loops $t_2$ that touch the orbifold locus $q_2 = q_3$. In the relative wedge, the angular coordinate $\varphi$ ranges from $\varphi = 0$ to $\varphi = \pi/3$.

Table 1: Character table for identity $e$ and generators $t_1$ and $t_2$ of $T_3 \sim D_\infty$. The generators $t_1$ and $t_2$ describe loops in $\mathcal{Q}_3$ that realize particle exchanges and their representations describe possible topological exchange statistics.

| irrep | $e$ | $t_1$ | $t_2$ | statistics |
|---|---|---|---|---|
| $(++)$ | 1 | $+1$ | $+1$ | bosons |
| $(--)$ | 1 | $-1$ | $-1$ | fermions |
| $(+-)$ | 1 | $+1$ | $-1$ | abelian traid anyon |
| $(-+)$ | 1 | $-1$ | $+1$ | abelian traid anyon |
| $0 < \alpha < \pi$ | 2 | 0 | 0 | non-abelian traid anyons |

The single two-dimensional representation of $S_3$ occurs as one of these representations of $T_3 \sim D_\infty$ corresponding to the specific angle of $\alpha = 2\pi/3$.

Single-component wave functions $\Psi(\mathbf{q})$ in the representation $(\tau_1 \tau_2 \ldots \tau_{N-1})$ are defined on $\mathbf{q} \in \mathcal{Q}_T$ and transform under an exchange path constructed from $M$ pairwise exchanges $\gamma = t_{i_M} \cdots t_{i_2} t_{i_1} \in T_N$ like

$$\hat{U}_\gamma \Psi(\mathbf{q}) = (\tau_{i_M} \cdots \tau_{i_2} \tau_{i_1}) \Psi(\mathbf{q}). \tag{C.3}$$

In the cases of bosons or fermions, this agrees with the transformation law (2) and $\Psi(\mathbf{q})$ can be lifted to a single-valued function on $\mathcal{X}_T = \mathbb{R}^N - \Delta_3$. However, if not all $\tau_i$ are the same, then the same permutation can be executed by topologically distinguishable exchange paths. Lifting the wave function for one of these abelian mixed representations to $\mathcal{X}_T$ so that every exchange $\gamma \in T_N$ is represented correctly results in a multi-valued function, as described in the main text.

# D   Haldane exclusion statistics

Above we have discussed topological exchange statistics, where indistinguishable particles are characterized by the transformations of their wave functions under continuous exchanges, leading to the possibility of braid anyons in 2D and traid anyons in 1D. Exclusion statistics on the other hand provides a completely different notion of anyons with fractional statistics. Originally proposed by Haldane [23,35], later used by Wu [75] and Polychronakos [24,25,76] to formulate a general theory of quantum statistical mechanics, exclusion statistics defines anyons as particles obeying a generalized Pauli exclusion principle, interpolating between bosons and fermions. Importantly, the theory makes no reference to the spatial dimension of the system, allowing for the existence of anyons not only in 2D (like it was historically assumed to be the case for exchange statistics), but also in 1D or 3D systems. Although the concepts of exclusion and exchange statistics in general do not give equivalent descriptions of fractional statistics, there are cases where both theories coincide, like for the quasiparticles in the fractional quantum Hall effect [23]. Various 1D systems exhibiting quasiparticles which obey a generalized Pauli principle are known in the literature. Famous examples include the Calogero-Sutherland model [77, 78], the Kundu model [42] and the Haldane-Shastry model [79]. Here we only give a brief review of the concept of exclusion statistics and refer to the literature for more details.

Consider $N_i$ identical particles of species $i$ in a system of $N$ particles in total. When we fix the coordinates of $N-1$ of these particles, the single-particle Hilbert space dimension of the remaining particle of species $i$ is denoted as $d_i\left(\{N_j\}\right)$, which generally depends on the particle numbers of all species $\{N_j\}$. We assume that the single-particle Hilbert space dimension $d_i$ is finite and extensive [23]. We now consider how $d_i$ changes when we add or subtract particles of different species ($\Delta N_j$) to the system. The change $\Delta d_i$ of the single-particle Hilbert space dimension is given by

$$\Delta d_i = -\sum_j \alpha_{ij}\Delta N_j\,, \tag{D.1}$$

where Haldane defined the statistical interaction $\alpha_{ij}$. In the case of bosons, the single-particle Hilbert space dimension does not change when particles of any species are added to the system and thus $\alpha_{ij}=0$. Fermions on the other hand obey the Pauli principle. This means, if a fermion of species $i$ is added to the system, the single-particle Hilbert space dimension $d_i$ gets reduced by 1, i.e. two fermions cannot occupy the same quantum state. In contrast, if a fermion of a different species $j$ is added, $d_i$ doesn't change. The statistical interaction of fermions is thus given by $\alpha_{ij}=\delta_{ij}$. Quasiparticles where the statistical interaction is different from the boson or fermion case are called anyons with fractional statistics. The special case where $\alpha_{ij}\neq 0$ for $i\neq j$ is also called mutual statistics. Building on Eq. D.1, Wu derived the occupation numbers of a gas of ideal anyons at finite temperature without mutual statistics ($\alpha_{ij}=\alpha\delta_{ij}$) [75]. For the mean occupation number $n_i$ at zero temperature Wu found

$$n_i = \begin{cases} 0\,, & \text{if } \epsilon_i > E_F\,, \\ 1/\alpha\,, & \text{if } \epsilon_i < E_F\,, \end{cases} \tag{D.2}$$

where $\epsilon_i$ is the single-particle energy of a particle in a state $i$ and $E_F$ is the Fermi energy. Moreover, at finite temperature $n_i \leq 1/\alpha$ holds. We thus have a simple interpretation of the statistical interaction $\alpha$: For example in the case of $\alpha = 0.5$, every quantum state with energy $\epsilon_i$ can be occupied by at most 2 anyons. In the ground state all of those states up to the Fermi energy are occupied by exactly 2 anyons. The same holds for other values $\alpha = 1/r$, where each state can hold up to $r$ particles.

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
