# Peer review of "Beyond braid statistics: Constructing a lattice model for anyons with exchange statistics intrinsic to one dimension"

_SciPost Physics, doi:SciPost Phys. 16, 086 (2024)_

## Round 2 · Referee Report · Anonymous (Referee 2) · 2023-12-12

Strengths

1) An extensive work of high professional quality. 2) Very well written. 3) Many useful figures. 4) Original ideas that may find future applications.

Report

This article is about particle statistics in one dimension, in
particular what the authors call traid statistics. They cite two of
their own earlier articles, refs. (16,17), where they define this
concept and discuss it in more detail. They present here the
continuum theory of traid statistics as a background for their
construction of a lattice model, which is their main purpose.

It is my opinion that the work is of high quality, and I recommend
publication. This conclusion comes with the qualification that I have
read the article somewhat selectively, and mainly the parts
concerning the continuum theory.

I have especially one comment, as follows, which I would like to see answered.

a) Theory, earlier version

The case of one dimension was discussed by Leinaas and Myrheim, ref. 1
in this article. The relative configuration space, ignoring the
centre of mass coordinate, was there identified as a space with
boundaries. Taking three identical particles as an example, the
relative space is the wedge depicted in Fig. 15(b) in the present
article. The solutions of the Schr{\"o}dinger equation must satisfy
boundary conditions designed to make the probability current normal to
the boundary vanish. The formula for the probability current depends
on the Hamiltonian, in standard quantum mechanics it is quadratic in
the wave function $\psi$, and a suitable boundary condition is the
linear condition

$\psi_n = \eta\psi,$

where $\psi_n$ is the normal derivative of the wave function, and $\eta$
is a real parameter. Here $\eta=0$ means bosons, $1/\eta=0$ means
fermions, but any intermediate positive or negative value of $\eta$ is
theoretically possible.

This mathematical approach is called in the present article the
manifold approach, because it is based on the configuration space
which is a manifold with a boundary, in one-to-one correspondence with
the physical configurations.

b) Theory, version presented here

An alternative approach is taken in the present article. The wave
functions are defined on $R^3$, the configuration space describing a
system of identical particles carrying arbitrary labels 1,2,3, and the
indistinguishability is manifested by identifications between
different sectors where the particles 1,2,3 appear in different orders
along the line, as illustrated in Fig. 13. A boundary between two
sectors is singular, because the identification implies that it has
only one side.

They call this the orbifold approach, as opposed to the manifold
approach, because the space where the wave functions are defined is an
orbifold and not a manifold, it contains singularities which are
manifolds of lower dimensions.

c) Difference between the two versions

It is surprising to me that the possibility of statistics intermediate
between bosons and fermions, as argued in ref. 1, is not recognized in
the orbifold approach. One would think that this is a case where
conclusions should not depend on the mathematical approach.

It is not immediately clear, however, how to translate from the
manifold to the orbifold formulation of the quantum theory. One
problem is that it has to be defined by convention what should happen
to the wave function $\psi$ along a curve crossing the boundary. The
necessary conditions are that the probability density $|\psi|^2$ has to
be symmetric across the boundary, and the probability current must
have a component normal to the boundary which is antisymmetric.

In the bosonic case, the obvious convention is that the wave function
is symmetric across the boundary, implying that the normal derivative
is zero on the boundary. Similarly in the fermionic case, the wave
function is naturally assumed to be antisymmetric across the boundary,
vanishing on the boundary in order to be continuous. In both cases,
the wave function and its derivative will be continuous along a
differentiable curve crossing the boundary. But it should be realized
that these rules for crossing the boundary are strictly speaking pure
conventions. It is quite possible to describe bosons by antisymmetric
wave functions and fermions by symmetric wave functions.

The question then is how to formulate an orbifold version of the
quantum theory with "$\eta$-statistics". It must be possible, but I do
not know how (or am too lazy) to do it.

d) Two more comments

They argue in addition that a strong three-body repulsion excluding
from the configuration space points where three or more particles
collide makes the one-dimensional traid anyons more similar to
two-dimensional braid anyons. It is clear that this similarity has
its limits. As seen in Fig. 13, a loop encircling the excluded point
in the three-particle case will have to pass through six boundaries
where two particles cross, and each crossing can at most result in a
change of sign of a one-component wave function. Therefore the loop
can not result in phase factors more general than +-1.

I wonder, by the way as a wild idea, whether it is possible to assume
that the crossing of the boundary results in a complex conjugation of
the wave function (possibly accompanied by the multiplication with a
fixed phase factor).

---

## Round 2 · Referee Report · Anonymous (Referee 1) · 2023-12-12

Strengths

1- The paper is very well written and clear. 2- The paper introduces traid-anyon statistics and many related concepts in a very pedagogical fashion.

Weaknesses

1- The proposed lattice Hamiltonian features all-to-all N-body couplings and is therefore quite unrealistic. 2- Numerical results are very qualitative. In particular, it is unclear which aspects are inherently topological (i.e. can be achieved only through traid-anyon statistics and not through local interactions in a bosonic Hamiltonian).

Report

After summarizing the conceptual points that lead to anyon braid statistics in 2+1D, the authors introduce anyon traid statistics: a relatively novel type of statistics that can be defined in 1+1D. The reasoning, although far from trivial, is explained very clearly and pedagogically for readers who are not experts on this topic, making the paper self-contained. Throughout the text, the authors introduce a lattice model for traid anyons and analyze it numerically. Finally, they study the continuum limit of the model and show that they recover results that had been previously derived in different ways.

In general, the lattice approach and the pedagogical style retained throughout the manuscript make the paper a great resource for other physicists who might want to approach this novel topic.

I think the manuscript is a valuable contribution that could foster new work on the topic from a broader community of condensed matter physicists. However, the numerical simulations and their interpretation seem rather confusing and it is often unclear what rationale led the authors to certain interpretations. Therefore I would recommend its publication only if the numerical study of the model is substantially improved.

In Sec. III, the authors summarize the construction of 1D lattice models designed to mimic braid anyon statistics. Proceeding along similar lines the authors are able to identify a lattice model with non-trivial traid statistics. Having a lattice Hamiltonian displaying this type of statics is potentially useful to both theoretical and experimental physicists, as the authors explain in their motivation (i)-(iii) on page 2.

However, I think that the model constructed here addresses motivation (i) and (iii) only partially: the proposed lattice models have non-local and even $N$-body interactions, where $N$ is the total number of particles (used in the hopping operator of the rightmost pair of particles). In the current implementation, this is needed to correctly reproduce the traid phases on the lattice. In this sense, the model does not seem realistic even for an artificial quantum system.

1- If the authors have in mind a specific setup in which Hamiltonian (14) can be efficiently realized, it would be very valuable if they could discuss it explicitly. Otherwise, I think the authors should be more open about this limitation and discuss it explicitly in the introduction and the conclusions.

In Sec. IV, the authors then proceed to numerically investigate the properties of traid anyons. This could be, in principle a strong point of the paper, since it is enabled by the lattice construction here presented for the first time. However, in its current form, this might be the weakest part of the paper. I am sympathetic towards the fact that the physics of these systems is new and good paradigms to interpret numerical data might be missing, but the current discussion does not provide much value.

One quantity studied by the authors is the chemical potential. However, the very notion of chemical potential appears to be ill-defined for these systems. Calling $E_0(N)$, the ground state energy of a system with $N$ particles, the authors define the chemical potential as $E_0(N)-E_0(N-1)$. However, the model is not fully specified by N, since every new particle introduces a sign choice in the traid group representation.

2- In what sense is the choice made by the author meaningful? Is there a way of relating the notion of chemical potential to physically measurable properties here to resolve this ambiguity? To remove this ambiguity, could the behavior of two-point functions be a better quantity to investigate?

3- I would expect the dependency with N not to be determined entirely by topological effects, but by interactions as well. Could the author comment on these effects?

4- Could the authors add an explanation of why the fermions and pseudofermions results are different? Naively, I would have thought that the two should coincide.

Secondly, the authors observe some approximate quantization of the eigenvalues of the one-particle density matrix, which the authors refer to as (near) integer-valued occupation numbers''. However, numerically, this seems to happen only for some choices of $N$ and $\tau_i$. In the current form, it is not clear if this(near) integer-valued occupation" is a coincidence that happens sometimes for very small system sizes or something physically meaningful.

5- What does "near" mean in this context? Is it intended that in some appropriate limit, the occupation numbers will tend to integer values? Otherwise, is there a sense for which the (near) quantization can be expected in many instances of $N$ and statistics $\tau_i$?

6- The authors further write "We again emphasize that this is a feature of the traid anyons, and it is strikingly different from bosons or fermions." While I agree that free bosons will have different eigenvalues, I expect that these values will depend on the details of the interactions as well. Is there a sense in which some features of the eigenvalue are purely "topological" and cannot be mimicked by a local interacting Hamiltonian for bosons?

7- I also wanted to ask why numerical DMRG results are limited to rather small sizes. I don't know if studying larger sizes would be beneficial here, but if there are issues with this numerical technique, it would be useful if the authors could comment on them.

Overall, I want to emphasize the manuscript is a valuable contribution. Its weakest point is the numerical section, where it is unclear how the results should be interpreted. Provided that this is substantially improved, I would recommend its publication in SciPost Physics.

Requested changes

I would ask the authors to address points 1-7 above and change the manuscript accordingly.

---

## Round 2 · Referee Report · Anonymous (Referee 3) · 2023-12-17

Strengths

  • The numerical indications of a possible fractional exclusion principle are intriguing.
  • The paper is well written.

Weaknesses

  • Any experimental realization of the proposed model appears out of reach.

Report

In the manuscript "Beyond braid statistics: Constructing a lattice model for anyons with exchange statistics intrinsic to one dimension”, Nagies et al. propose a 1D lattice model for so-called "traid" anyons and study its properties including the continuum limit. The authors first review exchange statistics in continuum models, discussing conventional statistics, braid statistics, and traid statistics, a concept introduced by some of the authors. The possibility of traid statistics exists for particles with a three-body hard-core constraint in 1D. The authors construct a lattice model for abelian traid anyons. The starting point is a bosonic model with number dependent Peierls phases, which provide the required geometric phases. This construction is in some sense analogous to the description of Abelian anyons in terms of flux tube -particle composites. The model itself is interesting and appears sound. That it contains a non-local many-particle string operator, however, will make any realization, be it with ultracold atoms in optical lattices or any other platform, challenging if not unrealistic. The authors also investigate the ground-sate properties of the model and find tentative indications for fractional exclusions statistics, including fractional Friedel oscillations, a step-wise behavior of the chemical potential as a function of the total particle number and near-integer occupations of the natural orbitals. The signatures the authors report are not unambiguous, but intriguing and in need for further investigation.

The manuscript is an the long side, but well written. It will not be of interest to a wide readership, but to the best of my judgement valued by experts.

In summary, I recommend it for publication in its present form.

---

## Round 2 · Referee Report · Anonymous (Referee 4) · 2023-12-21

Strengths

  1. Excellently written, pedagogical and self contained.
  2. Introduces an interesting lattice model that realises anyonic statistics in 1D in a 'natural' way, provides a very complete discussion of its basic properties, and also gives numerical evidence for some features.
  3. Also derives the appropriate continuum limit model and shows that it is consistent with previous results for certain anyonic wavefunctions.
  4. High quality, useful, figures.

Weaknesses

  1. The later subsections of section V are perhaps not quite as lucid as the earlier parts of the manuscript.
  2. It may be that the model can't be realised experimentally (though it is still interesting theoretically).

Report

In this excellently written, and highly readable manuscript the authors propose a lattice model that realises anyonic statistics of a certain type in 1D.
Models with unusual statistics are of fundamental interest, and this model is novel because it retains Galilean invariance, unlike previously introduced models of anyonic statistics in 1D. Furthermore, the mechanism behind the statistics is atypical because it invokes a three body hard core constraint.

The key results are the introduction of the lattice model, and its continuum counterpart, which open new research directions for those working on topological phases and many-body quantum physics in 1D.
The lattice model is also useful in that when written in terms of so-called traid-anyons it resembles the familiar bose-Hubbard model, which makes interpreting it somewhat more natural. The price of this simple form for the traid-anyons is that complicated non-local strings in the hopping terms for the underlying boson model become non-local commutation relations for the traid-anyons.
The introductory sections (I and II) are clear, pedagogical, well self-contained, and a model for other authors. The figures are of high standard and useful to the reader.

There are some queries I'd like the authors to address.

Requested changes

  1. Unless I misunderstand, as written the manuscript seems to consider particles on a line, not on a ring. The results in figs 7, 8 seem to confirm this. Are there any complications for periodic boundary conditions? If so, are there any topological implications for the thermodynamic limit?
  2. The authors refer to their model as quadratic. I wouldn't consider the Hubbard model quadratic due to the U interaction term which is quartic in creation/annihilation operators. Can the authors please clarify this?
  3. Two body and three body hard-core constraints and their implications are, understandably, discussed in detail. Is there any meaning to higher, n-body constraints? Might they lead to different physics, or possibly would they still lead to a similar continuum theory, in which there is an emergent two-body hard core constraint anyway?
  4. The authors refer to pseudo-fermions. It might be useful to explain a bit more why the traid-anyons with the ----... signature are only pseudo-fermions, and what the origin of the difference between fermions and pseudo-fermions in fig 9 is.
  5. I found the discussion of Fig 6. a bit confusing. The text describes processes that aren't shown in the figure, e.g. for 6a the text describes hopping to the right to create. a double occupied site, and then hopping off again. But the latter process isn't shown. In fact none of 6 a,b,c seem to show a particle swap, so I think either a new figure or new description is needed.
  6. The third line of Eq 20 doesn't seem relevant - possibly I'm missing something, but does this definition for $N_{jk}$ get used anywhere, as $sgn(i-j)$ is zero for $i=j$ anyway?
  7. I found the following sentence fragment on page 5 a bit hard to parse: "as they correspond to continuous transformations of strands that are not possible also in the presence of two-body hardcore interactions". Please consider rephrasing this.
  8. The difference between the behaviour of even and odd numbers of particles is reminiscent of the behaviour of the quantum Ising model in the fermionic field theory limit after J-W transformation. In that case the difference between even and odd numbers of particles means that both symmetric and antisymmetric boundary conditions have to be considered, and the spectrum separates into Ramond and Neveu-Schwarz sectors. Is something similar happening for the traid-anyon field theory?

Typos: 8. Page 7, "In comparison neither $\mathcal{T}$ for $\mathcal{P}$...", for -> nor? 9. Page 13, just above the start of section V. Something has gone wrong with the definition of $t_\alpha$, presumably the text in red was supposed to be removed?

---

## Round 3 · Referee Report · Anonymous (Referee 4) · 2024-2-19

Report

I'm very happy with the authors' responses to my queries and the (minor) alterations to the manuscript.

I believe the manuscript meets criterion 3 of the journal expectations: by developing a Hubbard like 1-D model of anyonic statistics that doesn't break Galilean invariance, the authors have opened up a significant new research pathway.
It also addresses a long standing stumbling block (criterion 2), by providing a useful model of anyonic statistics in 1D.

All the general acceptance criteria are met: this is an extremely well-written and complete manuscript.

---

## Round 3 · Referee Report · Anonymous (Referee 1) · 2024-2-27

Report

I am generally happy with the author's reply to my comments.
The discussion of the numerical results has greatly improved, especially in Subsection 4.3.
While the definition of the chemical potential still looks very arbitrary to me and the reasoning provided by the authors does not fully lift my objections, this is potentially the only weak point of an otherwise well-thought-out and carefully written paper.
I would therefore recommend publication of the manuscript in its current form.

---

## Round 3 · Author Response

We thank the four referees for their reports. All of them encouraged us towards publication in SciPost Physics, but they also made comments that substantially improved the paper. We have included a point by point response to all four referees below. We have also revised the arXiv pre-print to the SciPost format.

---

## Round 3 · List of Changes

REFEREE 1:

We thank the Referee 1 for evaluating our submission to SciPost Physics and we are happy about the positive feedback "The reasoning, although far from trivial, is explained very clearly and pedagogically for readers who are not experts on this topic, making the paper self-contained. [...] In general, the lattice approach and the pedagogical style retained throughout the manuscript make the paper a great resource for other physicists who might want to approach this novel topic." We appreciate that the referee believes that (pending revisions) our manuscript is on track for publication. We also thank the Referee for their constructive critical remarks, regarding especially the numerical simulation of the traid anyon lattice model, which we address in detail below and according to which we made revised the manuscript. (Please note, all references now refer to the new arXiv version.)

Request 1: If the authors have in mind a specific setup in which Hamiltonian (14) can be efficiently realized, it would be very valuable if they could discuss it explicitly. Otherwise, I think the authors should be more open about this limitation and discuss it explicitly in the introduction and the conclusions.

Response 1: Referee 1 observes correctly that we do not formulate an experimental implementation for our model in this article. As we emphasize in the text, the exchange phase depends on ordinality (i.e., the number of particles to the left or right of a given exchange). Ordinality is a non-local notion that depends on the $N$-body configuration space, although perhaps calling it an $N$-body interaction obscures its relative simplicity. We think there there may be several possible approaches to implementing ordinality-dependent phases in the lab, but it is too premature to include these speculations. So we have added new sentences to the manuscript to clarify that although the experimental implementation of traid anyons is a motivation, we do not actually propose an implementation in this article and that such an implementation would be more difficult than the recent implementation of the (braid)-anyon-Hubbard model.

In the introduction, we now write: "While we do not propose a specific implementation for our model, we do identify the necessary structure of the number-dependent tunneling phases than an implementation would require, and that due to the non-local multi-body interactions required, it will be more complicated than in the recently-realized anyon-Hubbard model [14]."

Also later, in the conclusions, we have added a comment: "It is an interesting task for future research to design experimentally accessible schemes for the implementation of this (or related) traid-anyon models, though such an implementation will have to address the challenge of effectively realizing the required non-local interactions."

Request 2: One quantity studied by the authors is the chemical potential.
However, the very notion of chemical potential appears to be ill-defined for these systems. Calling E0(N), the ground state energy of a system with
N particles, the authors define the chemical potential as E0(N)−E0(N−1). However, the model is not fully specified by N, since every new particle introduces a sign choice in the traid group representation. In what sense is the choice made by the author meaningful? Is there a way of relating the notion of chemical potential to physically measurable properties here to resolve this ambiguity? To remove this ambiguity, could the behavior of two-point functions be a better quantity to investigate?

Response 2: As Referee 1 notes, for each particle that is added, one has a choice between adding it "like a boson" symmetrically or "like a fermion" antisymmetrically, and so in that sense there is not a unique thermodynamic limit. However, when we restrict ourselves to the two alternating representations $(+-+-\cdots)$ and $(-+-+\cdots)$ (or more generally, to any representation with a repeating pattern of $\tau_i$, including also the case of bosons $\tau_i=1$, and fermions, $\tau_i=-1$), then we can speak of a well-defined thermodynamic limit.

In the revised manuscript, this issue is now pointed out more clearly. Moreover, we also discuss the reason for choosing this quantity in more detail. We have added the text:

"Note that this quantity is well defined only for a specific rule for how the sign $\tau_N$ is chosen, when the $N+1$st particle is added to the system. It is of interest, since it directly provides a perspective on the relation between exchange statistics and exclusion statistics (see Appendix~\ref{app:exclusion} below). Free bosons and free fermions (where by ``free'' we mean non-interacting) are the extreme cases. Any number of bosons can occupy the ground state, so each additional particle requires the same amount of energy, whereas each additional fermion must occupy the next higher single-particle energy eigenstate. Free particles with fractional exclusions statistics, where single-particle states can be occupied by a well-defined finite number of particles, would, in a similar fashion, give rise to a clear fingerprint in the behaviour of the chemical potential as a function of the total particle number. Below, we will, indeed, see traces of such behaviour, but also deviations from it, which will be attributed to the fact that, as a result of the number-dependent tunneling phases, even for $U=0$ the traid-anyon Hubbard model does not describe free particles."

Request 3: I would expect the dependency with N not to be determined entirely by topological effects, but by interactions as well. Could the author comment on these effects?

The Referee is absolutely right. A clean fingerprint of exclusion statistics is visible in the ground-state chemical potential only for free (i.e. non-interacting particles). It turns out, however, that even for $U=0$ the particles of the proposed traid-anyon Hubbard model are not free. This is also known from the braid-anyon-Hubbard model, where the effective interactions between the bosons have been analyzed; c.f. Ref.[30]. Nevertheless, we still see signatures in the dependence of the chemical potential with respect to the total particle number that reflect the expected behaviour, but slightly modified by interactions. In response to the Referee, we now discuss this point in more detail. See the new text cited in our response to point 2 above as well as the new sentence "This hints at an effectively induced attractive interaction, something previously noted for the braid-anyon-Hubbard model [30].", which we included later in the same section IV B.

Request 4: Could the authors add an explanation of why the fermions and pseudofermions results are different? Naively, I would have thought that the two should coincide.

Reply 4: We have added a sentence explaining this discrepancy, which is evident for higher densities. This occurs because pseudo-fermions act like bosons on the same site, and this effectively reduces the energy cost of adding pseudo-fermions to the system. In the revised manuscript, we now write in chapter IV C about pseudofermions (in view of the discrepancy with real fermions in Fig. 9):

"Their chemical potential agrees nicely with that of free fermions in the dilute regime (roughly below quarter filling or $N=7$), but as the density increases one can see a deviation because multiply-occupied sites are allowed for pseudo-fermions, unlike for true fermions."

Request 5: Secondly, the authors observe some approximate quantization of the eigenvalues of the one-particle density matrix, which the authors refer to as (near) integer-valued occupation numbers''. However, numerically, this seems to happen only for some choices of N and tau_i. In the current form, it is not clear if this "(near) integer-valued occupation" is a coincidence that happens sometimes for very small system sizes or something physically meaningful.

What does "near" mean in this context? Is it intended that in some appropriate limit, the occupation numbers will tend to integer values? Otherwise, is there a sense for which the (near) quantization can be expected in many instances of N and statistics tau_i?

Request 6: The authors further write "We again emphasize that this is a feature of the traid anyons, and it is strikingly different from bosons or fermions." While I agree that free bosons will have different eigenvalues, I expect that these values will depend on the details of the interactions as well. Is there a sense in which some features of the eigenvalue are purely "topological" and cannot be mimicked by a local interacting Hamiltonian for bosons?

Combined reply to 5 and 6: The motivation for this subsection on natural orbitals is again to try and probe the interplay among exchange statistics and exclusion statistics, by comparing the traid-anyon-Hubbard model to bosons and fermions. For $N$ free bosons, we would expect one SPDM eigenvector (i.e., the ground state) to have occupation $N$ and the rest to be $0$, whereas for free fermions there would be $N$ eigenvectors (i.e., natural orbitals) with occupation $1$, corresponding to the filled Fermi sea. For free bosons and fermions, the natural orbitals would be the single-particle states. In this context, integer values for occupation numbers are taken a signature of a system of "free" particles in its ground state.

For anyons obeying fractional exclusion statistics, we expect that single-particle orbitals can be occupied by several particles up to a maximum occupation (being 2 for the example of semions). For a system of free anyons with fractional exclusion statistics in its ground state, each orbital would be occupied by an integer number of particles, not exceeding the maximum possible occupation.

There are however discrepencies. We tested whether this was a finite-size effect (the answer was no), and as we describe in the text our best understanding is that this is due to the interaction effectively induced by the number-dependent phases on the lattice. But we admit that our results are not yet conclusive.

We have extensively modified the text of Sec. VI C, to address these questions and make it more clear why near-integer values for the expectations is physically motivated.

Request 7: I also wanted to ask why numerical DMRG results are limited to rather small sizes. I don't know if studying larger sizes would be beneficial here, but if there are issues with this numerical technique, it would be useful if the authors could comment on them.

Reply 7: The low-particle number, low-density regime is where we believe the connection to topological exchange statistics will be most evident. In this regime, exact diagonalization is sufficient to achieve numerical results. We added a clarifying note that only for the higher-particle numbers of Fig. 9 did we apply DMRG to calculate ground state energies.

A further motivation for studying this low filling-factor regime is the recent implementation of the braid-anyon-Hubbard model, which currently has two particles on eight sites. However, other experimental groups are seeking to look at the higher-density regime, and future studies with DMRG will be useful there.

Summary: We made substantial changes to Section IV, where the numerical results are presented, to address the Referee's concerns and demonstrate that the numerical results we present are critical to addressing the research goals of this article.
Further, we made clarifications in the introduction and conclusion addressing the challenging possibility of implementation.
* * *
REFEREE 2

We thank Referee 2 for the positive report. Further, we appreciate the question about comparing the Leinaas-Myrheim $\eta$-statistics for 1D anyons to the orbifold approach. One of the goals of creating and exploring the traid-anyon-Hubbard model is to understand the interplay of exchange statistics, exclusion statistics, and interactions, so the referee's question is most welcome, and we have several comments in response.

In the Leinaas-Myrheim approach, the $\eta$-parameter describes the boundary condition at the two-body coincidences. The parameter is a degree of freedom that remains after the constraint is applied that there is no probability current through the boundary. Alternatively, in the bosonic case, this parameter can be identified (up to a constant) with the two-body coupling constant for the Lieb-Liniger gas, and when Leinaas-Myrheim anyons are symmetrically extended to the configuration space $\mathbb{R}^N$, they have been called Lieb-Liniger anyons; c.f. Ref. [45]. The limiting case corresponds to the Tonks-Girardeau limit, c.f. Ref. [39], and the bosons are said to have been `fermionized', meaning that their probability density and energies are equivalent to fermions, although their momentum distribution is not.

The point of this digression is that the Leinaas-Myrheim approach is equivalent to imposing two-body interactions on bosons (or alternatively, a more complicated derivative delta-interaction on fermions, c.f. [40, 41]). This interaction has consequences for exclusion statistics (a la Haldane), but seems to be largely independent of exchange statistics. In fact in the Leinaas and Myrheim approach, fermions are equivalent to bosons, they just satisfy different boundary conditions.

However, for a manifold with boundary, as in the Leinaas and Myrheim approach, the fundamental group is trivial, there is no notion of exchange by a continuous adabiatic path, and there is no possibility for topological exchange statistics. For a much longer discussion, see Ref. [17]. In Section V.E. and Appendix C, we have sketched how, even in the absence of two-body interactions/boundary conditions, there can still be non-trivial statistics. As the referee notes, this does not mean that one cannot additionally impose $\eta$-type exclusion/interaction statistics on top of the topological structure of the orbifold. An earlier version of this paper included an appendix on this subject, but it was incomplete and far from the mainline of the article; we intend to perform just that extension in a future work. The exciting possibility that opens up in the orbifold approach is that there is a different $\eta$-parameter for each sequential pair, and the required representations of the traid group are non-abelian.

As for the intriguing suggestions in the final comments of the referee about possible extensions, we have thought a little about that. As Valiente points out in Ref. 24 (see comments after e.q. (105)), one challenge is making sure that generalizations preserve the gauge-independence of the interactions from the labels of the particles. If this accidentally introduces distinguishability to the particles, it can lead to spurious results. However, the possibility of looking for projective representations of the traid group (or the traid group combined with time reversal) would certainly be interesting to explore in more detail.
* * *
REFEREE 3

We thank Referee 3 for reviewing our work and are gratified by the recommendation in current form. However, we have made changes in response to other referees that we think address two of the criticisms in the first report.

First, in response to Referee 1 we have added a sentence to the introduction about possible realizations, pointing out that the non-local dependence of the phase will challenge an implementation. However, we think that their might ways of effectively realizing the required non-local coupling, for instance on the basis of correlated quantum states that obey non-local string order. Second, we made other revisions in response to Referee 1 that clarify the significance of our numerical results.
* * *
REFEREE 4

We thank Referee 4 for their comments and encouragement toward publication and the helpful comments and questions. We have responded to the referee's comments below and made the necessary changes to the paper:

Request 1: Unless I misunderstand, as written the manuscript seems to consider particles on a line, not on a ring. The results in figs 7, 8 seem to confirm this. Are there any complications for periodic boundary conditions? If so, are there any topological implications for the thermodynamic limit?

Response: The referee is correct; both the braid-anyon-model and our new traid-anyon-model are defined on a finite lattice with open boundary conditions. There exists work on the braid-anyon-Hubbard on a ring, with both twisted and periodic boundary conditions, including for the thermodynamic limit. However, the traid group on a ring does not have abelian representations besides bosons and fermions. This is explained in Ref. [17], but the point is that even if the Yang-Baxter relation is broken, the fact that particles can `go around the back' enforces that all particles have the same statistics in abelian representations. On a line (or interval), one has a well-defined notion of ordinality, which our model depends on to define the number-dependent phase, but this is lost on the ring. Therefore, we did not explore periodic or twisted boundary conditions for our model of abelian anyons.

Request 2: The authors refer to their model as quadratic. I wouldn't consider the Hubbard model quadratic due to the U interaction term which is quartic in creation/annihilation operators. Can the authors please clarify this?

Response: Thanks for catching this. In most cases we were intending to refer to the kinetic term of the Hamiltonian, or referring to the case when there was no on-site interaction, but the language was incorrect. We have gone through and fixed this.

Request 3: Two body and three body hard-core constraints and their implications are, understandably, discussed in detail. Is there any meaning to higher, n-body constraints? Might they lead to different physics, or possibly would they still lead to a similar continuum theory, in which there is an emergent two-body hard core constraint anyway?

Response: For an $N$-body constraint to have topological consequences, the defect it creates in configuration space must either make the space not path-connected or not simply-connected. The only $N$-body interaction that breaks the path-connected of space is two-body interactions in 1D. This is called a co-dimension 1 defect (like a line in a plane, or a plane in a 3D volume), because each two-body coincidence $x_i -x_j =0$ is one-dimensional. To make the configuration space not simply connected, the defect must have co-dimension 2, like a point in a plane or a line in a volume. There are only three $N$-body interactions that do this (c.f. Refs. [23,24]): two-body in 2D with constraint defects described by $x_i - x_j =0$ and $y_i - y_j =0$; three-body in 1D with constraints like $x_i - x_j =0$ and $x_j - x_k =0$; and a non-local 4-body interaction in 1D with constraints like $x_i - x_j =0$ and $x_k - x_l =0$. Dimensional analysis shows that this list really is exhaustive.

On a separate note, a two-body hard-core constraint in any dimension implies a three-body constrain, but only if all pairs of particles feel the constraint. In the case of the alternating representations of the traid group, the two-body constraint between sequential antisymmetric pairs is also enough to force the three-body constraint for the equivalent continuum bosonic model. However, for traid anyon representations with sequences of multiple consecutive $\tau_i = +1$, then the two-body constraints implied by the $\tau_i = -1$ for certain pairs are not sufficient to guaranteed three-body exclusion. This is mentioned in the text after Eq. 44.

Request 4: The authors refer to pseudo-fermions. It might be useful to explain a bit more why the traid-anyons with the ----... signature are only pseudo-fermions, and what the origin of the difference between fermions and pseudo-fermions in fig 9 is.

Referee 1 also commented on this, and We have added a comment to clarify.

Request 5: I found the discussion of Fig 6. a bit confusing. The text describes processes that aren't shown in the figure, e.g. for 6a the text describes hopping to the right to create. a double occupied site, and then hopping off again. But the latter process isn't shown. In fact none of 6 a,b,c seem to show a particle swap, so I think either a new figure or new description is needed.

Response: We have revised that paragraph in section III B to make it more clear.

Request 6: The third line of Eq 20 doesn't seem relevant - possibly I'm missing something, but does this definition for Njk get used anywhere, as sgn(i−j) is zero for i=j anyway?

Response: We thank the referee for pointing that out. We have changed Eq. (20) accordingly.

Request 7: I found the following sentence fragment on page 5 a bit hard to parse: "as they correspond to continuous transformations of strands that are not possible also in the presence of two-body hardcore interactions". Please consider rephrasing this.

Response: We thank the Referee for pointing that out. We have fixed the problem.

Request 8: The difference between the behaviour of even and odd numbers of particles is reminiscent of the behaviour of the quantum Ising model in the fermionic field theory limit after J-W transformation. In that case the difference between even and odd numbers of particles means that both symmetric and antisymmetric boundary conditions have to be considered, and the spectrum separates into Ramond and Neveu-Schwarz sectors. Is something similar happening for the traid-anyon field theory?

Response: This is a fascinating comment, and worthy of further investigation. We were previously unaware of the RNS construction, but it may have relevance, at least as an analogy. Like the two-body hard-core contact interaction in 2D, the three-body hard-core contact interaction in 1D is conformally invariant, which seems to be an ingredient necessary for the RNS construction to be relevant. Further (as emphasized by Referee 2 and alluded to somewhat in Sect. V.E.), the continuum version of traid anyons can be mapped to bosons or fermions with boundary conditions. We thank the referee for this idea, and will follow up, but do not think it is necessary to include in this already long paper.

Finally, thanks for catching a few typos, which have now been corrected.

---

## Editorial Decision

published